# Learning Collective Variables from BioEmu with Time-Lagged Generation

**Seonghyun Park**[1]   **Kiyoung Seong**[1]   **Soojung Yang**[2]
**Rafael Gómez-Bombarelli**[2]   **Sungsoo Ahn**[1][†]
[1]KAIST, [2] MIT
{hyun26, kyseong98, sungsoo.ahn}@kaist.ac.kr,
{soojungy, rafagb}@mit.edu

## Abstract

Molecular dynamics is crucial for understanding molecular systems, but its applicability is often limited by the vast timescales of rare events like protein folding. Enhanced sampling techniques overcome this by accelerating the simulation along key reaction pathways, which are defined by collective variables (CVs). However, identifying effective CVs that capture the slow, macroscopic dynamics of a system remains a major bottleneck. This work proposes a novel framework coined BioEmu-CV that learns these essential CVs automatically from BioEmu, a recently proposed foundation model for generating protein equilibrium samples. In particular, we re-purpose BioEmu to learn time-lagged generation conditioned on the learned CV, i.e., predict the distribution of molecular states after a certain amount of time. This training process promotes the CV to encode only the slow, long-term information while disregarding fast, random fluctuations. We validate our learned CV on fast-folding proteins with two key applications: (1) estimating free energy differences using on-the-fly probability enhanced sampling and (2) sampling transition paths with steered molecular dynamics. Our empirical study also serves as a new systematic and comprehensive benchmark for MLCVs on fast-folding proteins larger than Alanine Dipeptide.

## 1 Introduction

Many problems in biomolecules reduce to understanding the complex dynamics of molecular motion and conformational changes, such as estimating drug-target binding affinities (De Vivo et al., 2016; Abel et al., 2017) or the folding times of proteins (Piana et al., 2012; Spotte-Smith et al., 2022). Molecular dynamics (MD) simulation is one of the principal methods for studying molecular motion, describing microscopic dynamics by integrating differential equations with femtosecond ($10^{-15}$) time steps. However, events like protein folding often happen in a much larger timescale, from microseconds ($10^{-6}$) to milliseconds ($10^{-3}$). Due to this timescale gap, observing desired events in naive MD requires integrating over a computationally infeasible number of steps, which is considered unrealistic for most real-world problems.

To overcome this timescale limitation, various enhanced sampling techniques have been studied (Hénin et al., 2022). Metadynamics (Barducci et al., 2011) applies biasing force to the molecular systems to encourage transitions, while replica-exchange MD (Sugita & Okamoto, 1999) exchanges configurations between parallel simulations at different temperatures. Also, accelerated MD (Hamelberg et al., 2004) globally boosts the potential energy surface to overcome energy barriers. Their core idea is to add biasing forces throughout the simulation to guide exploration and transitions in the molecular state space, without breaking the equilibrium conditions. Importantly, these biasing forces are computed based on a low-dimensional representation, denoted as the *collective variables* (CVs). Therefore, CVs well encoding the slow degree of freedom will add biases to numerous seen CVs, resulting in exploration and transition to unseen and less visited states.

---

[†]Corresponding Author.

However, CVs have been mainly hand-crafted by domain expertise, e.g., selection of specific backbone dihedral angles for Alenine Dipeptide, which may miss slow modes and has been limited to small systems. To resolve this issue, several works have considered machine-learned CVs (MLCVs) from MD trajectory data. Supervised methods use ground-truth state definitions and pseudo-labels (Bonati et al., 2020; Trizio & Parrinello, 2021), while self-supervised methods rely on time-lagged data to encode the dynamics information (Bonati et al., 2021; Wehmeyer & Noé, 2018; Hernández et al., 2018). While prior works have shown promising results on small systems such as alanine dipeptide, they currently lack the ability to scale to larger systems such as proteins and lack a standardized, side-by-side systematic comparison.

In this work, we present a simple yet efficient framework for learning collective variables (CVs) from the latent representation of a molecular foundation model. Inspired by recent research extracting condition representations from text-to-image diffusion models (Zhang et al., 2023; Mou et al., 2024), we train an MLCV encoder to learn the latent representations in an existing molecular foundation model, BioEmu (Lewis et al., 2025). Building upon its capability of generating multiple protein conformations, we condition time-lagged data and use them as CVs for enhanced sampling techniques. Additionally, we benchmark prior MLCVs on three fast-folding proteins for two downstream tasks by enhanced sampling simulations along with extensive qualitative analysis.

To summarize, our contribution is as follows:

- We propose a framework for extracting collective variables from the time-lagged condition representations of a frozen foundation model.
- We extensively benchmark MLCVs with two downstream tasks for the slow degree of freedom: free energy difference estimation and transition path sampling for proteins in explicit water solvent simulations.

## 2 BACKGROUND

**Molecular dynamics (MD).** MD simulations model the time evolution of molecular systems through atomic coordinates and velocities. An illustrative MD is the underdamped Langevin dynamics (Bussi & Parrinello, 2007) as follows:

$$\mathrm{d}x_t = v_t\mathrm{d}t, \quad \mathrm{d}v_t = \frac{-\nabla U(x_t)}{m}\mathrm{d}t - \gamma v_t\mathrm{d}t + \sqrt{\tfrac{2\gamma k_B T}{m}}\mathrm{d}W_t\,,$$

where $x_t$ and $v_t$ are atomic positions and velocities, $m$ is the mass, $U(x_t)$ the potential energy, and $-\nabla U(x_t)$ the force. The parameters $\gamma$, $k_B$, $T$, and $W_t$ denote the friction coefficient, Boltzmann constant, temperature, and Brownian motion, respectively. While MD simulations provide atomic-level insights into dynamic process, it remains limited by timescales. The high energy barriers between states makes rare events difficult to observe such as transition paths, often requiring unrealistic long MD simulations (Valsson et al., 2016).

**Enhanced sampling and collective variables (CVs).** To overcome the timescale limitations of the standard MD simulations, enhanced sampling methods introduce biasing forces or potentials to accelerate transitions. Popular approaches include metadynamics (Barducci et al., 2011, MTD), well-tempered metadynamics (Barducci et al., 2008, WTMD), replica-exchange molecular dynamics (Hukushima & Nemoto, 1996, REMD), and on-the-fly probability enhanced sampling (Invernizzi & Parrinello, 2020, OPES). These methods typically rely on a low-dimensional descriptors known as *collective variables* (CVs), which serve as a biasing coordinate in the simulation (Bonati et al., 2023). Formally, the CVs, denoted by $c$, is represented as a set of functions of atomic position $x$:

$$c(x) = [\xi_1(x), \ldots, \xi_d(x)],$$

where $d \ll 3N$ is the number of CVs. In this work, we fix the dimensionality to one for simplicity and visibility. CVs are designed to capture the system's slow degree of freedom, often chosen based on domain knowledge, such as specific backbone dihedral angles or inter-residue contact distances (Piana & Laio, 2007). However, handcrafted CVs remain mostly limited to small systems (Pietrucci & Laio, 2009; Noé & Clementi, 2017).

**Machine-learned collective variables (MLCVs).** Recent methods employ machine learning to automatically identify CVs beyond handcrafted features. Supervised approaches such as

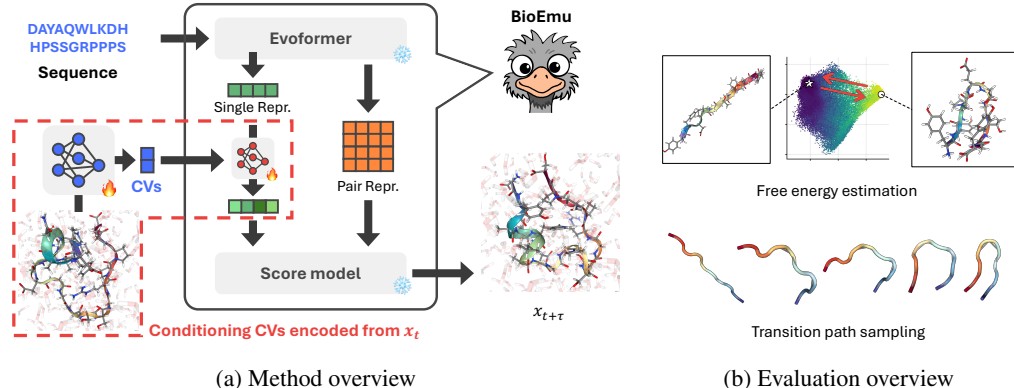

(a) Method overview  (b) Evaluation overview

Figure 1: **Overview of our framework and evaluation simulation. (Left)** We train an encoder on top of the conditions of a frozen molecular foundation model to learn collective variables (CVs) for protein, highlighted in red dotted lines, on top of a frozen pre-trained BioEmu. **(Right)** Two downstream tasks for the slow degree of freedom, free energy estimation and transition path sampling.

DeepLDA (Bonati et al., 2020) and DeepTDA (Trizio & Parrinello, 2021) learn CVs through discriminant analysis. However, they require predefined binary state labels, where one needs to specify a representative folded state and choose an RMSD threshold for classification. In contrast, self-supervised and time-lagged methods instead leverage dynamical information from trajectories. DeepTICA (Bonati et al., 2021) applies time-lagged independent component analysis (Molgedey & Schuster, 1994, TICA) as a loss for encoded representations. The time-lagged autoencoder (Wehmeyer & Noé, 2018, TAE) and variational dynamics encoder (Hernández et al., 2018, VDE) reconstructs future configurations $x_{t+\tau}$ from the present configuration $x_t$, deriving latent representations as CVs using autoencoders (Rumelhart et al., 1985) and variational autoencoders (Kingma & Welling, 2014), respectively. However, existing methods mostly focus on small systems such as alanine dipeptide, and systematic comparisons across approaches are absent.

## 3 METHOD

**Overview.** We aim to learn an encoder that outputs a low-dimensional vector representing the slow degree of freedom in molecules, known as collective variables (CVs). Inspired by recent works on conditioning frozen foundation models (Zhang et al., 2023; Mou et al., 2024), we re-purpose the BioEmu model. As shown in Figure 1a, instead of using it for its original purpose of generating protein conformation ensemble, we add an encoder to extract its low dimensional representation to serve as CV, which we call BIOEMU-CV. This CV learns the slow dynamics by approximating the molecular dynamics (MD) propagator with a conditional generative model, which generates a subsequent state from a current state and a time lag similar to Hernández et al. (2018).

**Task formulation.** Given a protein configuration represented by atomic coordinates in $x \in \mathbb{R}^{N \times 3}$ where $N$ is the number of atoms, our goal is to learn an encoder $f_\theta$ that maps the high-dimensional structure into a *low-dimensional representation* $c = f_\theta(x) \in \mathbb{R}^d$ with $d \ll 3N$. This representation, the CVs, should qualify three criteria for the use of enhanced samplings: (i) being low-dimensional, (ii) capturing the system's slow degree of freedom, and (iii) discriminating the folded and unfolded states in protein (Fu et al., 2024). We fit the first criteria by using a one-dimensional CV, and evaluate the meta-stable state discrimination with secondary structures which are highly correlated to the folded states. For the second and most important criterion, we evaluate MLCVs with two downstream tasks requiring to encode the slow degree of freedom: (i) free energy estimation and (ii) transition path sampling, each by different enhanced sampling techniques.

**Biomolecular Emulator (BioEmu).** BioEmu (Lewis et al., 2025) generates the conformational ensemble at full-atom resolution unlike structure prediction models such as AlphaFold (Jumper et al., 2021), which return a single low-energy conformation given a protein sequence. We focus on the capability to generate protein conformation ensemble, and re-purpose it for learning collective variables (CVs). Formally, given an amino acid sequence $A = [A_1, A_2, \ldots, A_n]$ of length $n$, BioEmu

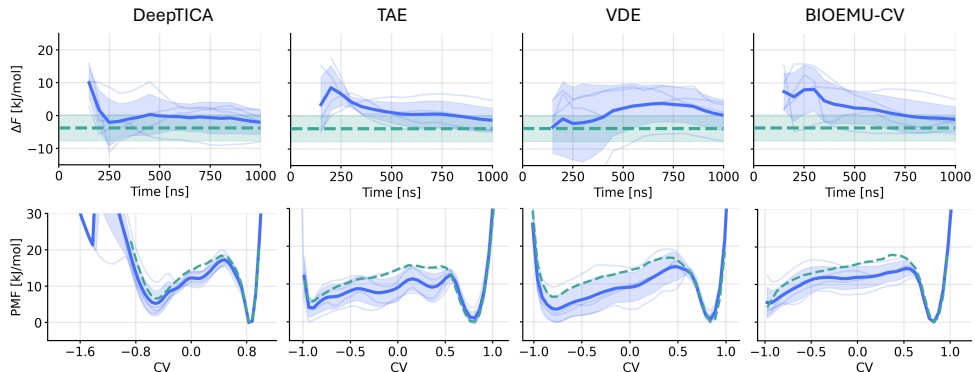

Figure 2: **Free energy (top) and PMF (bottom) estimation from 1 $\mu s$ OPES simulations for Chignolin.** We average over four simulations. Green dotted lines indicate the reference value, and blue lines refer to the free energy difference during the OPES simulations. Solid lines refer to the mean, and shaded areas are the standard deviation. DeepTICA shows negative values beyond -1 in PMF even when normalized, and falls short of accurately capturing the reference PMF.

is a sequence-conditioned denoising diffusion model $g_\phi(x|A)$ to sample from the equilibrium distribution $p(x|A)$. To be specific, the protein sequence encoder of the BioEmu produces two representations from the amino acid token with Evoformer (Jumper et al., 2021): the single representations $h := \{h_i\}_{i=1}^N$ and the pair representations $z := \{z_{ij}\}_{i,j=1}^N$ where $i, j$ refer to the amino acid index. Based on these two representations, the score model $g_\phi$ generates $\alpha$-carbon $(C_\alpha)$ coordinates and residue orientations for the protein backbone. Side chains are then reconstructed using the hpacker package (Visani et al., 2024), followed by short molecular dynamics (MD) simulations for energy relaxation with OpenMM (Eastman et al., 2017).

**Extracting CVs from foundation model.** Recent advances in generative models demonstrate that additional condition encoders with lightweight adapters can be trained on top of unconditional generative models for external guidance (Zhang et al., 2023; Mou et al., 2024). Inspired by this line of research, we propose a framework for using a low-dimensional encoder output as collective variables (CVs); an overview is shown in Figure 1. We train an encoder $f_\theta$ that maps a configuration $x$ at time $t$ to a low-dimensional vector, i.e., $c_t = f_\theta(x_t) \in \mathbb{R}^d$. Then, we fuse the encoded representation to the single representation $h$ using a small MLP to preserve dimensionality to the score model and keep the adapter lightweight (Zhang et al., 2023). The conditioned representation $h_t = \text{MLP}(h, c_t)$ and pair representation $z$ are passed to the score model to generate protein conformations.

**Learning CVs from time-lagged generation.** With this condition pathway in place, we encode CVs of the current conformation but guide the model to generate the time-lagged conformation $x_{t+\tau}$, i.e., the score model generates the time-lagged state $x_{t+\tau}$ from the CVs $c_t$ with a time lag $\tau$. Intuitively, the CV $c_t$ compresses the information that is shared across $x_t$ and $x_{t+\tau}$, i.e., the slow degree of freedom, while disregarding fast and randomly fluctuating information that is present in the current state $x_t$ but not in the time-lagged state $x_{t+\tau}$. Note that our training objective shares the motivation with VDE (Hernández et al., 2018), but we use a scalable diffusion model architecture and do not need the auto-correlation loss.

To keep the training lightweight, we freeze parameters $\phi$ of BioEmu, and update the encoder and the conditioning MLP via the denoising score-matching objective (Song et al., 2020; Yim et al., 2021):

$$\mathcal{L}(x_t, x_{t+\tau}, A) = \mathbb{E}_{s \sim \mathcal{U}[0,1]} \left[ \lambda_s \left\| \nabla \log p_{s|0}\left(x_{t+\tau}^{(s)} | x_{t+\tau}^{(0)}, x_t, A\right) - g_\phi(s, h_t, z) \right\|^2 \right], \quad (1)$$

where $s$ is the diffusion time, $p_{s|0}$ is the density of $x_{t+\tau}^{(s)}$ given $x_{t+\tau}^{(0)}, x_t, A, \lambda_s > 0$ the time scheduling weights, and $g_\phi$ the BioEmu's score network. This time-lagged reconstruction forces $c_t$ to capture the slow degree of freedom by incorporating future time-lagged conformation $x_{t+\tau}$ from the trajectory data, thereby producing CVs suitable for enhanced sampling.

Table 1: **Quantitative results of 1 $\mu$s OPES simulations for three fast-folding proteins in explicit water solvent.** We report the reference free energy difference ($\Delta F_{\text{ref}}$) from the data, the average free energy difference ($\Delta F$) from simulations, the absolute error between two energy values ($|\Delta F_{\text{ref}} - \Delta F|$), and the potential of mean force (PMF) MAE. Results are averaged over four simulations for Chignolin, and three simulations for Trp-cage and BBA. We mark not applicable (N/A) for CVs that fail at meta-stable state discrimination or show a different sign with the reference value.

| Molecule | Method | $\Delta F_{\text{ref}}$ | $\Delta F$ | $|\Delta F_{\text{ref}} - \Delta F|$ ($\downarrow$) | PMF MAE ($\downarrow$) |
|---|---|---|---|---|---|
| Chignolin | DeepTICA | -3.73 | $-2.02 \pm 3.65$ | 1.71 | $2.64 \pm 3.80$ |
| | TAE | -3.79 | $-1.26 \pm 3.69$ | 2.53 | $3.15 \pm 2.81$ |
| | VDE | -17.24 | $0.24 \pm 5.00$ | N/A | $4.09 \pm 3.20$ |
| | BIOEMU-CV | -3.71 | $-3.19 \pm 3.97$ | 0.52 | $3.07 \pm 2.53$ |
| Trp-cage | DeepTICA | 3.70 | $6.53 \pm 7.31$ | 2.73 | $8.94 \pm 7.43$ |
| | TAE | -1.45 | $8.74 \pm 1.65$ | N/A | $9.32 \pm 3.91$ |
| | VDE | 0.07 | N/A | N/A | N/A |
| | BIOEMU-CV | 4.15 | $5.97 \pm 3.01$ | 1.82 | $6.86 \pm 4.38$ |
| BBA | DeepTICA | 2.76 | $13.95 \pm 13.28$ | 11.19 | $10.51 \pm 5.85$ |
| | TAE | -2.88 | $-5.46 \pm 4.42$ | N/A | $9.97 \pm 3.68$ |
| | VDE | -2.74 | N/A | N/A | N/A |
| | BIOEMU-CV | 2.77 | $9.99 \pm 5.43$ | 7.22 | $8.34 \pm 7.46$ |

## 4 EXPERIMENT RESULTS

**Evaluation setup.** We consider three fast-folding protein in explicit water solvent from Lindorff-Larsen et al. (2011); Chignolin, Trp-cage, and BBA. For more details on protein data, refer to Appendix A. We quantitatively evaluate MLCVs with two downstream tasks that require encoding the slow degree of freedom: (i) free energy difference estimation, and (ii) transition path sampling. We use on-the-fly probability enhanced simulation (Invernizzi & Parrinello, 2020, OPES) and CV-steered MD simulation (Izrailev et al., 1999; Fiorin et al., 2013b, SMD) for each task, respectively. Furthermore, we extensively investigate the interpretability of CVs with respect to input descriptors with the sensitivity analysis, and analyze meta-stable state discrimination with secondary structures known to be present in the folded states following Fu et al. (2024). For ablation experiment results regarding architecture components, please refer to Appendix B.

**Baselines.** We compare our method against self-supervised CV learning methods: DeepTICA (Bonati et al., 2021), time-lagged autoencoder (Wehmeyer & Noé, 2018, TAE), and variational dynamics encoder (Hernández et al., 2018, VDE). Since protein types and simulation configuration vary significantly by methods and lack systematic comparison, we train from scratch on identical data and time-lag with baselines using the mlcolvar package (Bonati et al., 2023). We use pairwise $C_\alpha$ distances for inputs to ensure rotation and translation invariance following prior works (Trizio & Parrinello, 2021; Bonati et al., 2021), and fix CVs dimensionality to one. All MLCVs are normalized to the range $[-1, 1]$ on the full DESRES trajectory after training for simplicity and visibility, with the sign assigning the folded state to positive. For more details on baselines, refer to Appendix C.

### 4.1 FREE ENERGY DIFFERENCE ESTIMATION

We first quantify whether MLCVs encode the slow degree of freedom by estimating the free energy difference $\Delta F$, with on-the-fly probability enhanced simulations in explicit water solvent.

**On-the-fly probability enhanced simulations.** We run four independent 1 $\mu$s on-the-fly probability enhanced sampling (Invernizzi & Parrinello, 2020, OPES) simulations initialized from the folded state. OPES iteratively reconstructs the target probability along the CVs at a pre-defined interval, adding bias that continuously drives transitions between the folded and unfolded states. This enables the estimation of the folding free-energy difference and the potential of mean force (PMF).

**Evaluation criteria.** Following Yang et al. (2024), we measure (i) the convergence of the estimated reaction free energy of folding ($\Delta F$), and (ii) the quality of the potential of mean force (PMF).

Table 2: **Quantitative results of steered molecular dynamics on three fast-folding proteins in explicit water solvent.** RMSD and THP are averaged over paths obtained from 16 SMD simulations, while max energy ($E_{TS}$) is averaged over paths hitting the target meta-stable state. $k$ is the force constant, which is a scaling factor of the harmonic bias potential used in SMD. Best results are highlighted in **bold** and second in underline. We mark not applicable (N/A) for CVs that fail at state discrimination and trajectories not arriving at the target meta-stable state.

| Molecule | Method | $k$ | RMSD ($\downarrow$) Å | THP ($\uparrow$) % | $E_{TS}$ ($\downarrow$) kJ/mol |
|---|---|---|---|---|---|
| Chignolin | DeepTICA | 10000 | $2.45_{\pm 0.86}$ | 37.5 | $-81102.41_{\pm 521.27}$ |
| | TAE | 10000 | $\underline{1.95}_{\pm 0.72}$ | $\underline{43.8}$ | $-81914.87_{\pm 114.30}$ |
| | VDE | 10000 | $2.08_{\pm 0.56}$ | $\underline{43.8}$ | $\underline{-82026.62}_{\pm 77.63}$ |
| | BIOEMU-CV | 10000 | $\mathbf{1.20}_{\pm 0.33}$ | **100.0** | $\mathbf{-82055.15}_{\pm 98.48}$ |
| Trp-cage | DeepTICA | 20000 | $\underline{2.37}_{\pm 0.47}$ | **31.2** | $\underline{-63611.88}_{\pm 57.49}$ |
| | TAE | 20000 | $2.75_{\pm 0.35}$ | 0.0 | N/A |
| | VDE | N/A | N/A | N/A | N/A |
| | BIOEMU-CV | 20000 | $\mathbf{2.31}_{\pm 0.52}$ | **31.2** | $\mathbf{-63787.51}_{\pm 31.23}$ |
| BBA | DeepTICA | 50000 | $\underline{2.67}_{\pm 0.37}$ | $\underline{18.8}$ | $\underline{-130418.50}_{\pm 477.68}$ |
| | TAE | 50000 | $4.83_{\pm 0.42}$ | 0.0 | N/A |
| | VDE | N/A | N/A | N/A | N/A |
| | BIOEMU-CV | 50000 | $\mathbf{2.05}_{\pm 0.24}$ | **93.8** | $\mathbf{-131315.59}_{\pm 116.23}$ |

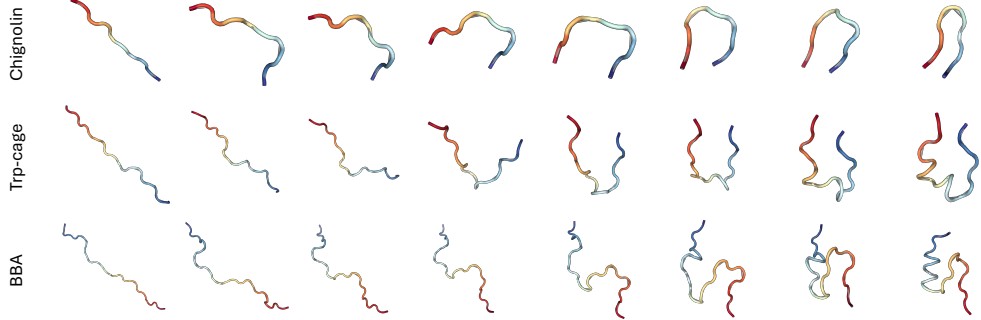

Figure 3: **3D Visualization of transition paths.** The sampled folding pathways of Chignolin, Trp-cage, and BBA by steered MD with BIOEMU-CV. We visualize only the $C_\alpha$ for simplicity.

We compute the two meta-stable state basins, folded and unfolded states, by dividing the CV range in half as in prior work. The reference values for each protein are computed from MLCVs of the full DESRES trajectory with the multi-state Bennett acceptance ratio analysis (Shirts & Chodera, 2008, MBAR). In short, CVs well encoding the slow degree of freedom are expected to have similar distributions between the 1 $\mu s$ OPES simulation and the long naive reference $\sim$100 $\mu s$ DESRES simulation. Note that we exclude results on MLCVs that fail on the basic criterion of discriminating between the folded and unfolded states; see more detail in Section 4.4. Since inadequate CVs produce non-converged biases and misleading PMFs, OPES cannot reliably drive transition, and its PMFs are meaningless. Results are averaged over four runs for Chignolin and three runs for Trp-cage and BBA, with one outlier excluded from the original four following Yang et al. (2024). For more details on the simulation, evaluation, and outlier criterion, refer to Appendix C.

**Results.** In Table 1, we report the reference free energy difference ($\Delta F_{\text{ref}}$), free energy difference averaged over simulations ($\Delta F$), and the potential mean force (PMF) MAE. In Figure 2, we additionally plot the free energy difference throughout the OPES simulations. For the full results, please refer to Appendix D. In the following, we interpret the results in both quantitative and qualitative aspects. While most MLCVs fairly converge for Chignolin and work for larger proteins, VDE fails to scale to large proteins. Additionally, TAE appears to converge for Trp-cage, but only because it samples the unfolded state. As shown in Figure 14, the folded state is barely visited, resulting

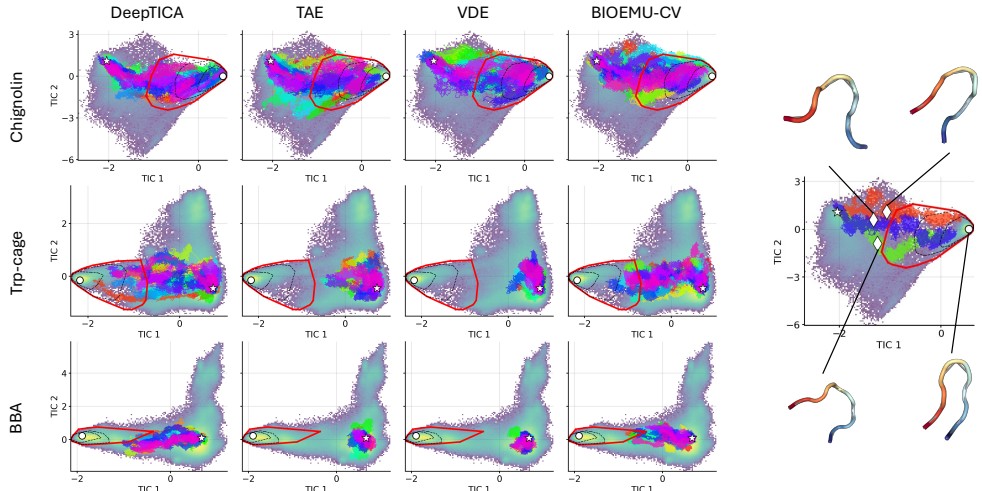

Figure 4: **Qualitative results on transition path sampling. (Left)** Transition paths from MLCV-steered MD projected onto TICA coordinates. The white star and circle each refer to the representative unfolded and folded state. The red convex hulls indicate the folded state regions based on RMSD cutoffs from the representative folded state with cut off 2, 2, and 2.5 Å for Chignolin, Trp-cage, and BBA, respectively. **(Right)** Diverse intermediate states for Chignolin transition paths using BIOEMU-CV steered MD, with white diamonds representing step 2,400 on TICA coordinates.

in convergence at an incorrect $\Delta F$, with the wrong sign relative to the reference value. Similarly, DeepTICA samples the folded state with high deviation even though the outlier was removed, resulting in fluctuation in the PMF. We have clarified more interpretations in detail in Appendix D.

## 4.2 TRANSITION PATH SAMPLING

We also evaluate the slow degree of freedom in CVs by sampling transition paths of the fast-folding proteins, with CV-steered molecular dynamics simulations in explicit water solvent.

**Steered molecular dynamics.** We sample transition paths with steered molecular dynamics (Izrailev et al., 1999; Fiorin et al., 2013b, SMD). SMD is an enhanced sampling technique that steers the protein conformation towards a pre-defined target state with a time-dependent biasing force. CVs that well encode the slow degree of freedom would produce low-energy transition paths when used with SMD. For each protein, we do 16 NVT simulations for 500 ps using the Langevin Integrator with 1 fs time step. For more detail about SMD, refer to Appendix C.

**Evaluation criteria.** To evaluate the transition paths from SMD simulations, we first define the target state as the local minimum on the potential energy surface corresponding to the folded state. We then evaluate the transition paths using three metrics from Seong et al. (2025): (i) $C_\alpha$-RMSD between the final state of a path and the target state, (ii) the target hit percentage (THP) of sampled paths, and (iii) the transition state energy ($E_{TS}$), i.e., the maximum energy of states of a transition path that hits the target meta-stable state. For THP, a transition path is considered to hit the target meta-stable state if its final state is within 2 Å $C_\alpha$-RMSD from the target state, regarding the state definition in Lindorff-Larsen et al. (2011). Again, for the transition path sampling task, any MLCVs that failed to discriminate between the folded and unfolded states of the protein were excluded.

**Results.** In Table 2, BIOEMU-CV shows the best results in hitting the target meta-stable states while achieving low transition state energies. Additionally in Figure 3, we visualize the $C_\alpha$ atoms in the folding process under BIOEMU-CV steered MD for qualitative assessment. For the visualization of baseline CV-steered MD, refer to Appendix D. In Figure 4, we also visualize the sampled transition paths of three fast-folding proteins from SMD simulations, by projecting them onto time-lagged independent component analysis (Molgedey & Schuster, 1994, TICA) made from the full DESRES trajectory. We find that transition paths from our MLCVs show better target reaching and follow the region with a low PMF value defined on two time-lagged independent components.

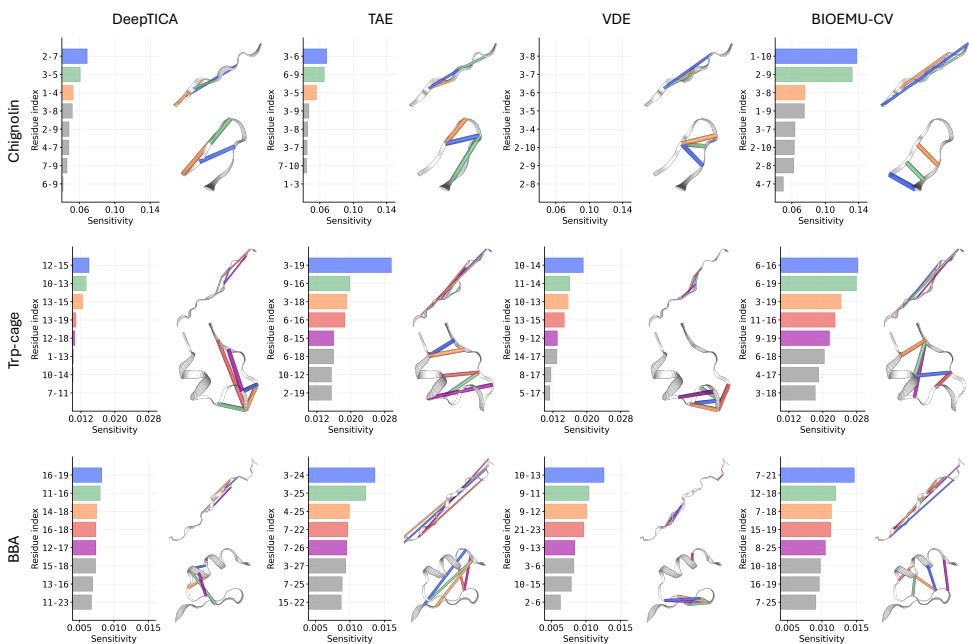

Figure 5: **MLCV sensitivity to $C_\alpha$-wise distances.** We plot the top $C_\alpha$-wise distances with the highest sensitivity for each MLCV, where the $x$ and $y$ axes each denote the residue index for input distances and the corresponding sensitivity value. For each sensitivity plot, we visualize the top sensitive distances in the unfolded and folded states, with colors highlighted in the sensitivity plot.

## 4.3 INTERPRETABILITY OF CVS

Beyond enhanced sampling tasks, we further assess the physical interpretability of MLCVs.

**Sensitivity analysis.** We evaluate the interpretability of MLCVs by examining how each MLCV changes to their input descriptors, i.e., the sensitivity to the $C_\alpha$-wise distance (Bonati et al., 2020; Trizio & Parrinello, 2021). To be specific, we use the sensitivity_analysis function from the mlcolvar library (Bonati et al., 2023), which computes the sensitivity by the gradients of each MLCV with respect to every input feature. We aggregated the values with the mean of the absolute values, and visualize the top-most sensitive distances in the folded state and unfolded state.

**Results.** As shown in Figure 5, BIOEMU-CV consistently assigns high sensitivity on distances that strongly discriminate the folded and unfolded state, typically long-range contacts spanning the secondary structure for folding. For example in Chignolin, BIOEMU-CV is most sensitive to the distance TYP1-TYR10 and also sensitive to ASP3-TRY8, where a hydrogen bond is observed at folding (Yang et al., 2024). In contrast, DeepTICA and VDE often emphasize distances that provide weaker structural discrimination, e.g., DeepTICA is sensitive to the distances of ASP3-GLU5 and TYR1-PRO4 which does not differ much between the folded and unfolded state. Overall, these results indicate that BIOEMU-CV not only captures the slow dynamical modes required for enhanced sampling, but also learn physically meaningful structural relationships.

## 4.4 META-STABLE STATE DISCRIMINATION ANALYSIS

Finally, we extensively analyze whether MLCVs clearly discriminate folded and unfolded states, following the criteria of Fu et al. (2024). To be specific, we test whether the folded and unfolded state ensemble occupy distinct ranges of MLCVs and report their distribution statistics across proteins. In case of Chignolin, we cross-validate against standard descriptors, e.g., the committor function and native hydrogen bond numbers. We also probe sensitivity to structural motifs by analyzing the secondary structure elements involved in the folded state, e.g., $\alpha$-helix and $\beta$-sheet.

Table 3: **State discrimination analysis of MLCVs.** We report the average MLCVs for the folded and unfolded states obtained from the full DESRES trajectory. VDE fails to separate the folded and unfolded states on Trp-cage and BBA, both showing positive ranges.

| Method | Chignolin | | Trp-cage | | BBA | |
|---|---|---|---|---|---|---|
| | Folded | Unfolded | Folded | Unfolded | Folded | Unfolded |
| DeepTICA | $0.85_{\pm 0.06}$ | $-0.49_{\pm 0.11}$ | $0.70_{\pm 0.12}$ | $-0.74_{\pm 0.01}$ | $0.75_{\pm 0.07}$ | $-0.51_{\pm 0.05}$ |
| TAE | $0.78_{\pm 0.07}$ | $-0.82_{\pm 0.13}$ | $0.94_{\pm 0.03}$ | $-0.95_{\pm 0.02}$ | $0.40_{\pm 0.08}$ | $-0.90_{\pm 0.05}$ |
| VDE | $0.84_{\pm 0.07}$ | $-0.74_{\pm 0.13}$ | $1.00_{\pm 0.00}$ | $0.87_{\pm 0.13}$ | $1.00_{\pm 0.00}$ | $1.00_{\pm 0.00}$ |
| BioEmu-CV | $0.82_{\pm 0.06}$ | $-0.94_{\pm 0.05}$ | $0.96_{\pm 0.03}$ | $-0.90_{\pm 0.03}$ | $0.94_{\pm 0.04}$ | $-0.93_{\pm 0.05}$ |

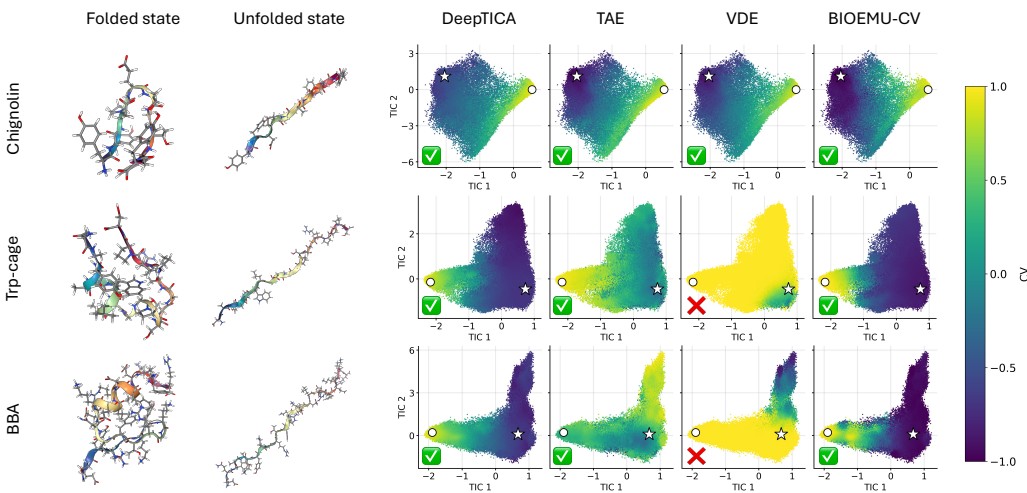

Figure 6: **(Left)** Visualizations of the folded and unfolded states for three protein. **(Right)** TICA projections of the full DESRES trajectory, colored by each MLCVs. The folded and unfolded states are marked by a white circle and a white star, respectively. All MLCVs are normalized to the range $[-1, 1]$, with signs flipped such that folded state corresponds to $+1$ for consistency and visibility. As the protein size increases, prior methods struggle with state discrimination, for instance, VDE.

**Meta-stable state discrimination.** First, we verify whether the CVs truly distinguish the folded and unfolded states of proteins. In Figure 6, we color each TICA coordinates with its MLCVs. We use a time lag of $\tau = 1000$ for BBA and $\tau = 10$ otherwise for TICA plots. While most methods distinguish the folded and unfolded states, prior works tend to fail at BBA. In Table 3, we report the average MLCVs for the folded and unfolded states gathered from the full DESRES trajectory. Noticeably, VDE shows similar values for Trp-cage and BBA. We plot the detailed distribution of MLCVs for the folded and unfolded states in Appendix D, where VDE fails at meta-stable state discrimination showing almost identical results between MLCVs of the folded and unfolded states.

**Chignolin analysis.** We also analyze CVs for Chignolin with well-known descriptors, the committor function. The committor function provides a quantitative measure of progress along the folding pathway on a scale from $[0, 1]$. We use the committor function from Kang et al. (2024), estimated in a data-driven manner. In Table 4, we report the Pearson correlation between the committor function and MLCVs. DeepTICA shows relatively weak correlation with the committor values, whereas other methods demonstrate stronger alignment. For more details on Chignolin descriptors, refer to Appendix C.

Table 4: **Pearson correlation** between MLCVs and the Chignolin committor function.

| Method | Pearson corr. |
|---|---|
| DeepTICA | 0.682 |
| TAE | 0.744 |
| VDE | 0.778 |
| BioEmu-CV | 0.748 |

**Secondary structure analysis.** Finally, we analyze the correlation between CVs and the secondary structure of proteins. As seen in Table 5, secondary structures are mostly observed in the

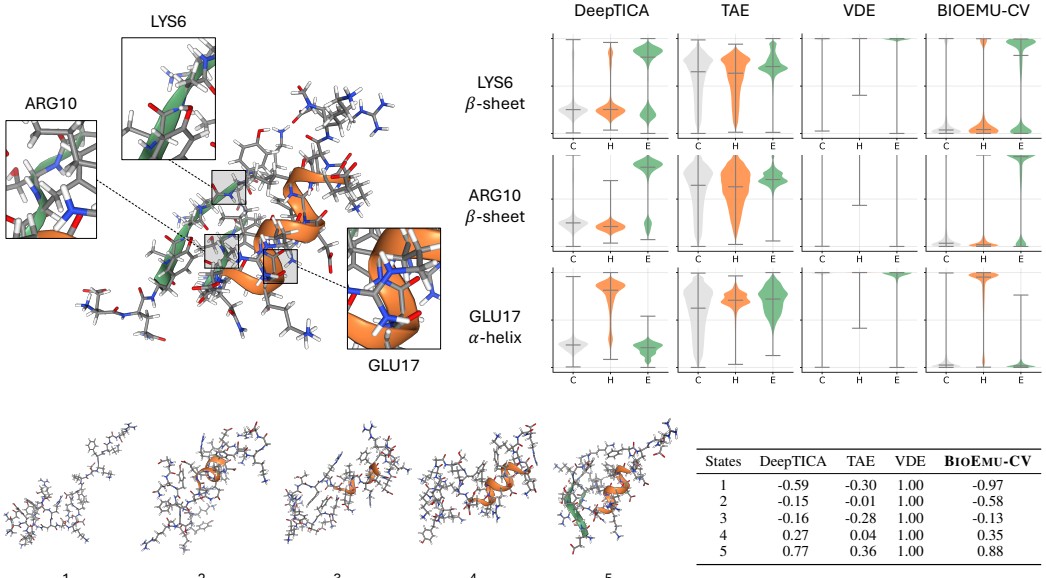

Figure 7: **Secondary structure analysis on BBA. (Top)** Folded-state visualization of the BBA protein, with $\alpha$-helix and $\beta$-sheet highlighted in orange and green, respectively. We also plot the MLCVs distribution for the residues included in the secondary structures, for residues showing specific secondary structures in the folded state. Letters C, H, and E each correspond to coil (irregular elements), $\alpha$-helix, and $\beta$-sheet secondary structures. TAE and VDE fail to assign distinct values to helix and sheet structures. **(Bottom)** Visualizations and MLCVs on states in the DESRES trajectory having partial secondary structures. Residues are colored according to the DSSP value. DeepTICA and BIOEMU-CV align with the secondary structures.

folded state clearly distinguished from the unfolded states (Lindorff-Larsen et al., 2011), therefore CVs should capture their presence to describe the folding dynamics (Ahalawat & Mondal, 2018).

We compute the residue-level secondary structure using the *dictionary of secondary structures in proteins* (Kabsch & Sander, 1983; McGibbon et al., 2015, DSSP), which classifies structures based on hydrogen-bond patterns. In Figure 7, we plot the MLCV distribution against the known secondary structures in BBA. While the folded states of BBA contain one $\alpha$-helix and two $\beta$-sheets (Lindorff-Larsen et al., 2011), TAE and VDE encode nearly identical CVs regardless of the secondary structures in contrast to DeepTICA and BIOEMU-CV successfully separating them.

Table 5: **Average fraction of residues forming the secondary structure in unfolded states**. Unfolded refers to the fraction of residues regardless of type.

| Protein | $\alpha$-helix | $\beta$-sheet | Unfolded |
|---|---|---|---|
| Chignolin | 0.00 | 0.00 | 0.00 |
| Trp-cage | 0.02 | 0.01 | 0.03 |
| BBA | 0.06 | 0.04 | 0.08 |

## 5 CONCLUSION

We present BIOEMU-CV, a lightweight framework that learns collective variables (CVs) from time-lagged conditioning signals on a frozen BioEmu foundation model. Given the current structure, we derive a low-dimensional CV and inject it into BioEmu's single representation, where only this conditioning path is trained with a denoising score-matching objective, leaving the backbone untouched. We systematically evaluate MLCVs on three fast-folding proteins for two downstream tasks, estimating free-energy differences using OPES simulations and transition-path sampling using CV-steered MD. Our experiments show that BIOEMU-CV successfully identifies important macroscopic movements related to the slow degree of freedom, and our qualitative evaluation of state discrimination corroborates this.

ETHICS STATEMENT

Our work introduces a method to accelerate the study of protein dynamics using the generative model. This work aims to advance scientific understanding of bimolecular mechanisms for disease analysis and drug discovery. We believe the potential for the misuse of this research is low.

REPRODUCIBILITY STATEMENT

All protein trajectory data used in this study are publicly available from the D. E. Shaw Research (DESRES) database. The detailed description of the dataset can be found in Lindorff-Larsen et al. (2011). We provided a brief explanation of the dataset in Appendix A.2. Implementation details for our method and all baselines, including model and training hyperparameters, are specified in Appendix C.2. Furthermore, the experimental setups for our downstream evaluation tasks are provided in the appendix; Appendix C.4 contains the configurations for the on-the-fly probability enhanced sampling (OPES) simulations, and Appendix C.6 outlines the setup for the steered molecular dynamics (SMD) simulations. For reproduction, we provide our code for *training & simulation*, and *simulation environment set up*.

ACKNOWLEDGMENTS

This work was supported by the National Research Foundation of Korea(NRF) grant funded by the Ministry of Science and ICT(MSIT) (No. RS-2022-NR072184), Korea Health Industry Development Institute(KHIDI) grant funded by the Ministry of Health & Welfare(MOHW) (No. N0425208, Developing a Highly Multimodal (Drug, Protein, Gene, Cell Imaging, Literature) Foundation Model for ADMET Property Prediction), the Institute for Information & communications Technology Planning & Evaluation(IITP)grant funded by the Korea government(MSIT) (RS-2019-II190075, Artificial Intelligence Graduate School Program(KAIST)), and the Institute of Information & Communications Technology Planning & Evaluation(IITP) grant funded by the Korea government(MSIT) (RS-2025-02304967, AI Star Fellowship(KAIST)).

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

# A    DATA DETAILS

## A.1    PROTEIN DATA

Proteins are a sequence of amino acid building blocks, each represented by one letter. Below, we provide statistics on the proteins we used in this work. Protein size ranges from 10 to 35 residues, 166 to 504 atoms. C The PDB ID in the Table 6 refers to the discovered experimental structures in the PDB with the smallest error. In case of Chignolin, it is closest to the NMR structure of CLN025 reported by Honda et al. (2008). Trp-cage corresponds to the K8A mutant of the thermostable Trp-cage variant TC10b in Barua et al. (2008), and BBA, i.e., short for beta-beta-alpha, corresponds to the FDS-EY peptide in Sarisky & Mayo (2001).

Table 6: **Details on three proteins** used in this work, selected among the DESRES fast folding proteins.

| Protein | PDB ID | # of Residues | # of Atoms | # of CA pairs | Sequence |
|---|---|---|---|---|---|
| Chignolin | CLN025 | 10 | 166 | 45 | YYDPETGTWY |
| Trp-cage | 2JOF | 20 | 272 | 190 | DAYAQWLKDHHPSSGRPPPS |
| BBA | 1FME | 28 | 504 | 378 | LSDEDFKAVFGMTRSAFANLPLWXQQHLXKEKGLF |

## A.2    DESRES TRAJECTORY DATASET

Lindorff-Larsen et al. (2011) provides twelve fast-folding proteins. Among them, we chose three proteins to test MLCVs. For proteins given with multiple simulations, we sample from the longest length simulation to use as the data. In other words, we use the simulation of length 223 $\mu s$ for the BBA protein. In Table 7, we denote the simulation time, average folding time, simulation temperature, and the cubic box size of the simulations. Since DESRES simulations are recorded in 0.2 picoseconds intervals and reach millions of frames, we chose to randomly sample 50,000 frames as in Lewis et al. (2025).

Table 7: **Simulation details** of three DESRES fast-folding proteins.

| Protein | Simulation time ($\mu s$) | Avg. folding time ($\mu s$) | Temperature (K) | Cubic box (Å) |
|---|---|---|---|---|
| Chignolin | 106 | 0.6 | 340 | 40 |
| Trp-cage | 208 | 14 | 290 | 37 |
| BBA | 223, 102 | 2.8 | 325 | 47 |

## A.3    TIME-LAGGED INDEPENDENT COMPONENT ANALYSIS

Time-lagged independent component analysis (Molgedey & Schuster, 1994, TICA) is a linear transformation method commonly used for dimensionality reduction, first used for molecular dynamics in Naritomi & Fuchigami (2011). In short, it maps the given data to the slow process for a given time-lag $\tau$, by finding the coordinates of maximal auto correlation between time-lagged data pairs. Therefore, TICA, combined with a long MD trajectory, will likely capture the slow degree of freedom well. In this work, we use the TICA model in pyemma (Scherer et al., 2015). All TICA plots in the paper, e.g., Figure 4 and Figure 6, are made with the full DESRES trajectory with a log norm applied. We use $C_\alpha$ pairwise distances for the input descriptors, and apply the switching function to obtain the contact $s_{ij}$ in the case of Chignolin as follows:

$$s_{ij} = \frac{1 - (r_{ij}/r_0)^n}{1 - (r_{ij}/r_0)^m},$$

where $r_{ij}$ refers to the distances between $C_\alpha$ atoms $i$ and $j$, $r_0$=0.8nm, $n = 6$, and $m = 12$. Intuitively, the contacts $s_{ij}$ are a continuous version of the coordinate numbers.

# B  ABLATION EXPERIMENTS RESULTS

In this section, we present various ablation experiments on BIOEMU-CV.

## B.1  TIME-LAGGED CONDITIONING AND FIXED BIOEMU WEIGHTS

Here, we conduct ablation experiment on not using time-lagged conditions and unfreezing BioEmu's parameters, with OPES simulations and Steered MD.

Table 8: **Ablation experiments for the components of BIOEMU-CV in OPES and steered MD simulations.** Time-lag indicates whether the model is trained to generated a time-lagged target conformation, and freezing indicates whether BioEmu's parameters are kept fixed during the training of the encoder. Our current design choice shows high performance for SMD results.

| Components | | OPES simulations | | | | Steered MD | | |
|---|---|---|---|---|---|---|---|---|
| Time-lag | Freezing | $\Delta F_{\text{ref}}$ | $\Delta F$ | $|\Delta F_{\text{ref}} - \Delta F|$ ($\downarrow$) | PMF MAE ($\downarrow$) | RMSD ($\downarrow$) Å | THP ($\uparrow$) % | $E_{TS}$ ($\downarrow$) kJ/mol |
| ✓ | ✓ | -3.71 | $-3.19_{\pm 3.97}$ | 0.52 | $3.07_{\pm 2.53}$ | $\mathbf{1.20}_{\pm \mathbf{0.33}}$ | **100.0** | $-82055.15_{\pm 98.48}$ |
| ✗ | ✓ | -3.68 | $-5.78_{\pm 3.20}$ | 2.10 | $1.41_{\pm 1.56}$ | $1.57_{\pm 0.36}$ | 81.3 | $-82084.68_{\pm 62.86}$ |
| ✓ | ✗ | -4.47 | $-3.25_{\pm 0.81}$ | 1.22 | $3.53_{\pm 3.73}$ | $1.62_{\pm 0.31}$ | **100.0** | $-82076.42_{\pm 98.20}$ |

**Time-lagged conditioning.** We input a conformation $x_t$ to the MLCV encoder, and condition the BioEmu to generate $x_t$ instead of using time-lagged data $x_{t+\tau}$. Formally, we are testing the denoising score-matching objective modified from Equation (1) as follows:

$$\mathcal{L}(x_t, A) = \mathbb{E}_{s \sim \mathcal{U}[0,1]} \left[ \lambda_s \left\| \nabla \log p_{s|0}\left(x_t^{(s)} | x_t^{(0)}, x_t, A\right) - g_\phi(s, h_t, z) \right\|^2 \right].$$

In Table 8, one can see that time-lagged conditioning results in a better performance for both downstream tasks, i.e., OPES and steered MD simulations. Intuitively, time-lagged data will inject dynamic information into the MLCVs, resulting enrich representations.

**Unfrozen BioEmu weights.** We also test whether unfreezing the BioEmu parameters would improve performance, while BIOEMU-CV only trains the parameters of the MLCV encoder. In Table 8, MLCV trained with a frozen BioEmu results in a lower RMSD in SMD and better OPES results. Since the performance is slightly better in unfrozen cases, keeping the lightweight training scheme is reasonable.

## B.2 ENCODER SIZES AND PLACEMENT

Table 9: **Quantitative results of steered MD for different encoder sizes.** The performance remains largely unchanged across the encoder size.

| Param. | Repr. | RMSD ($\downarrow$) Å | THP ($\uparrow$) % | $E_{TS}$ ($\downarrow$) kJ/mol |
|---|---|---|---|---|
| 48K | single | $\mathbf{1.20}_{\pm\,\mathbf{0.33}}$ | **100.0** | $-82055.15_{\pm\ 98.48}$ |
| 1.27M | single | $1.50_{\pm\,0.30}$ | 100.0 | $-82071.13_{\pm\,100.08}$ |
| 196K | single | $\underline{1.41}_{\pm\,0.32}$ | 93.8 | $-82049.40_{\pm\ 98.52}$ |
| 48K | pair | $1.79_{\pm\,0.45}$ | 56.3 | $\mathbf{-82088.33}_{\pm\,\mathbf{76.61}}$ |
| 48K | single, pair | $1.66_{\pm\,0.56}$ | 68.8 | $\underline{-82086.61}_{\pm\,74.80}$ |

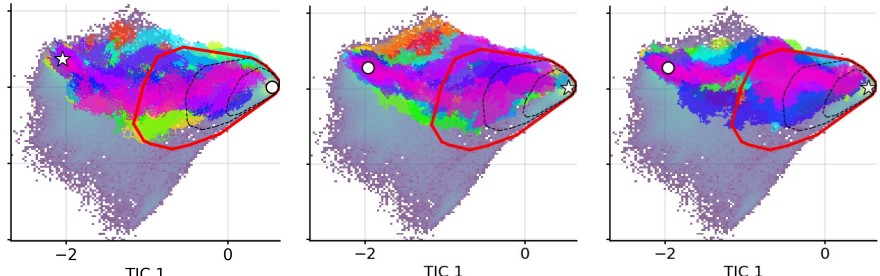

Figure 8: **Qualitative results of steered MD for different encoder sizes.** From left to right, each visualizes the paths from the steered MD from the encoder with 48K, 196K, and 1.27M parameters.

**Encoder size.** We test 48K, 196K, 1.27M parameters with layers {2, 4, 8} and hidden dimension {100, 200, 400} for Chignolin, where the main paper encoder corresponds to the 48K parameter setting. All MLCVs were mixed with $C_\alpha$ RMSD CVs at a 5:5 ratio. In Figure 8 and Table 9, performance remains largely unchanged across the encoder size. Since the encoder is inferred millions of times during the OPES simulation, keeping a relatively smaller size would be practical.

**Conditioning placement.** Additionally, we test whether on how selecting the conditioning representation affects on the performance. While BIOEMU-CV conditions the single representation as $h_t = \mathrm{MLP}(h, c_t)$, we additionally test conditioning the pair representation, i.e., $z_t = \mathrm{MLP}(z, c_t)$ and both the single and pair representation. In Table 9, RMSD and THP both fall behind the case of only conditioning the single representation, while the energy shows little improvement.

### B.3 STEERED MD WITHOUT MIXING CVS

Table 10: **Ablation experiments of steered molecular dynamics on three fast-folding proteins in explicit water solvent, without $C_\alpha$-RMSD CVs.** We mark not applicable (N/A) for CVs that fail at state discrimination and trajectories not arriving at the target meta-stable state.

| Molecule | $k$ | Method | RMSD ($\downarrow$) Å | THP ($\uparrow$) % | $E_{TS}$ ($\downarrow$) kJ/mol |
|---|---|---|---|---|---|
| Chignolin | 1000 | DeepTICA | $7.24_{\pm 1.49}$ | 0.0 | N/A |
| | | TAE | $5.72_{\pm 1.94}$ | 6.2 | $-81951.53_{\pm 0.00}$ |
| | | VDE | N/A | N/A | N/A |
| | | BIOEMU-CV | $7.82_{\pm 1.47}$ | 0.0 | N/A |
| | 2000 | DeepTICA | $7.01_{\pm 1.71}$ | 0.0 | N/A |
| | | TAE | $6.39_{\pm 1.74}$ | 0.0 | N/A |
| | | VDE | N/A | N/A | N/A |
| | | BIOEMU-CV | $6.09_{\pm 2.21}$ | 6.2 | $-82023.05_{\pm 0.00}$ |
| Trp-cage | 2000 | DeepTICA | $2.23_{\pm 1.23}$ | 0.0 | N/A |
| | | TAE | $8.62_{\pm 3.25}$ | 0.0 | N/A |
| | | VDE | N/A | N/A | N/A |
| | | BIOEMU-CV | $12.17_{\pm 1.58}$ | 0.0 | N/A |
| | 5000 | DeepTICA | $10.78_{\pm 1.15}$ | 0.0 | N/A |
| | | TAE | $7.22_{\pm 1.71}$ | 0.0 | N/A |
| | | VDE | N/A | N/A | N/A |
| | | BIOEMU-CV | $7.14_{\pm 1.52}$ | 0.0 | N/A |
| BBA | 50000 | DeepTICA | $7.39_{\pm 2.71}$ | 0.0 | N/A |
| | | TAE | $9.87_{\pm 1.59}$ | 0.0 | N/A |
| | | VDE | N/A | N/A | N/A |
| | | BIOEMU-CV | $6.03_{\pm 2.56}$ | 0.0 | N/A |
| | 100000 | DeepTICA | N/A | N/A | N/A |
| | | TAE | $10.20_{\pm 1.59}$ | 0.0 | N/A |
| | | VDE | N/A | N/A | N/A |
| | | BIOEMU-CV | $6.64_{\pm 1.95}$ | 0.0 | N/A |

**Steered MD without $C_\alpha$-RMSD CVs.** Here, we report the performance of steered MD using MLCVs without $C_\alpha$ RMSD. In Table 10, MLCV steered MD mostly fails to reach the target. The force constant $k$ was set to the maximum value with simulations not exploding, resulting in different $k$ values compared to Table 2. Due to this, we have incorporated $C_\alpha$ RMSD for all MLCVs in Table 2.

## C SIMULATION AND EVALUATION DETAILS

### C.1 QUALITATIVE VERIFICATION

**CLN025 descriptors.** CLN025 is known to form several hydrogen bonds at the folded state (Yang et al., 2024). The criteria for hydrogen bonds are (i) the donor-acceptor distance being smaller than 0.35 $nm$, and (ii) the angle formed by the donor, acceptor, and hydrogen being bigger than 110°. The donor acceptor atom list is as follows:

1. TYR1 N, TYR10 OT1
2. TYR1 N, TYR10 OXT
3. ASP3 N, TRY8 O
4. THR6 OG1, ASP3 O
5. THR6 N, ASP3 OD1
6. THR6 N, ASP3 OD2
7. GLY7 N, ASP3 O
8. TYR10 N, TYR1 O

**Committor function.** The committor function $q(x)$ is the probability that a trajectory initiated from a configuration $x$ reaches the folded state $B$ before the unfolded state $A$. It is the solution to the backward Kolmogorov equation with the boundary conditions $q(x) = 0$ for $x \in A$ and $q(x) = 1$ for $x \in B$. We use the committor function from Kang et al. (2024) to evaluate its correlation with MLCVs. In their approach, solving the Kolmogorov equation is reformulated as minimizing a variational functional. To this end, they parameterize the committor function as a neural network and train it using a self-consistent iterative procedure to optimize the functional.

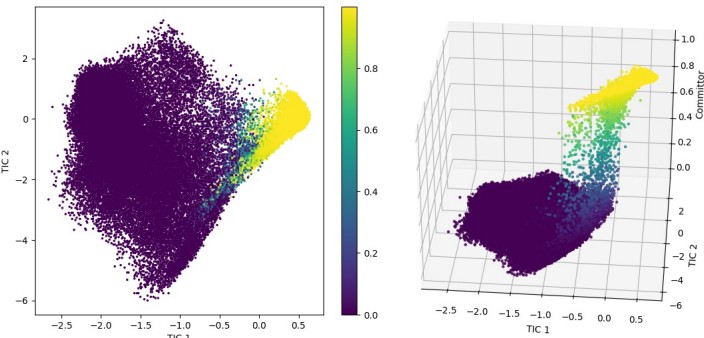

Figure 9: **TICA coordinates colored with committor values.**

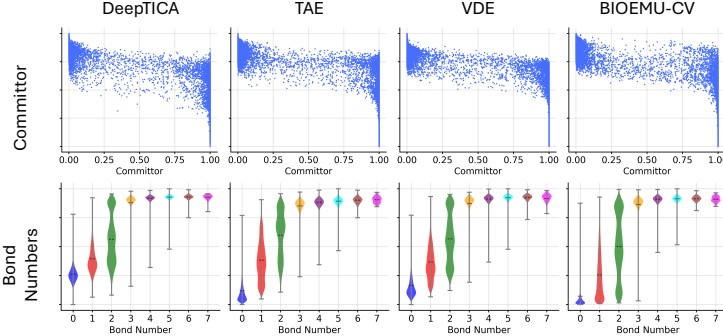

Figure 10: **Correlation to known descriptors for Chignolin. (Top)** Scatter plot of correlation to the committor function. **(Bottom)** Violin plot of correlation to the number of hydrogen bonds.

**Secondary structure.** Protein secondary structures are local spatial structures made from the backbone excluding the side chain, key structures observed in the folded states. The two most common structures are the $\alpha$-helix and $\beta$-sheet shown in Figure 7, i.e., spiral coil-like structure and flattened zig-zag-shaped structure, respectively. Secondary structures are formally defined by the pattern of hydrogen bonds between the hydrogen atoms in the amino part and the oxygen atoms in the carboxyl. In this work, we use the *dictionary of secondary structures proteins* (Kabsch & Sander, 1983, DSSP) for the definition of secondary structures, implemented in the MDtraj library (McGibbon et al., 2015). Note that secondary structures are assigned for each residue, and three or more consecutive residues with the same hydrogen bond are usually considered a full secondary structure. For proteins simulated in Lindorff-Larsen et al. (2011), secondary structures are rarely observed in the unfolded state as stated in Table 5. Additionally, in Table 11, we list the percentage of secondary structures throughout the whole DESRES trajectory for each residue and protein. Three classes of DSSP: C, E, and H each correspond to irregular elements, $\beta$-sheets, and $\alpha$-helix. While DDSP provides a detailed classification with eight classes, we selected the simplified one for this work.

Table 11: **DSSP class percentage of each residue** in Chignolin, Trp-cage, and BBA.

| DSSP | 0 | 1 | 2 | 3 | 4 | 5 | 6 | 7 | 8 | 9 |
|---|---|---|---|---|---|---|---|---|---|---|
| C | 100.0 | 25.5 | 53.8 | 99.2 | 98.6 | 98.4 | 98.5 | 52.7 | 24.7 | 100.0 |
| E | 0.0 | 74.5 | 46.2 | 0.0 | 0.2 | 0.1 | 0.1 | 46.3 | 74.6 | 0.0 |
| H | 0.0 | 0.0 | 0.0 | 0.8 | 1.2 | 1.5 | 1.5 | 1.0 | 0.6 | 0.0 |

| Trp-cage | 0 | 1 | 2 | 3 | 4 | 5 | 6 | 7 | 8 | 9 |
|---|---|---|---|---|---|---|---|---|---|---|
| C | 100.0 | 85.7 | 75.1 | 70.9 | 67.1 | 67.8 | 72.3 | 78.2 | 88.8 | 97.7 |
| E | 0.0 | 0.3 | 1.4 | 1.2 | 3.0 | 4.0 | 2.7 | 1.3 | 0.5 | 0.1 |
| H | 0.0 | 14.0 | 23.6 | 27.9 | 29.8 | 28.3 | 25.0 | 20.5 | 10.7 | 2.2 |

| Trp-cage | 10 | 11 | 12 | 13 | 14 | 15 | 16 | 17 | 18 | 19 |
|---|---|---|---|---|---|---|---|---|---|---|
| C | 83.6 | 80.9 | 75.7 | 80.1 | 95.6 | 94.6 | 100.0 | 100.0 | 99.8 | 100.0 |
| E | 1.7 | 2.1 | 4.6 | 2.4 | 0.6 | 5.4 | 0.0 | 0.0 | 0.2 | 0.0 |
| H | 14.7 | 17.0 | 19.7 | 17.5 | 3.8 | 0.0 | 0.0 | 0.0 | 0.0 | 0.0 |

| BBA | 0 | 1 | 2 | 3 | 4 | 5 | 6 | 7 | 8 | 9 |
|---|---|---|---|---|---|---|---|---|---|---|
| C | 100.0 | 96.4 | 79.1 | 75.5 | 68.1 | 64.6 | 70.1 | 89.2 | 93.7 | 73.9 |
| E | 0.0 | 3.0 | 9.5 | 10.7 | 16.0 | 21.0 | 23.1 | 5.1 | 1.3 | 20.2 |
| H | 0.0 | 0.6 | 11.4 | 13.8 | 15.9 | 14.4 | 6.8 | 5.6 | 5.0 | 5.9 |

| BBA | 10 | 11 | 12 | 13 | 14 | 15 | 16 | 17 | 18 | 19 |
|---|---|---|---|---|---|---|---|---|---|---|
| C | 70.8 | 75.5 | 91.0 | 94.6 | 71.6 | 67.5 | 67.3 | 67.3 | 67.4 | 67.7 |
| E | 23.8 | 19.4 | 5.9 | 3.8 | 3.0 | 4.6 | 4.3 | 2.9 | 2.7 | 2.6 |
| H | 5.5 | 5.1 | 3.2 | 1.5 | 25.4 | 27.9 | 28.4 | 29.8 | 30.0 | 29.7 |

| BBA | 20 | 21 | 22 | 23 | 24 | 25 | 26 | 27 | | |
|---|---|---|---|---|---|---|---|---|---|---|
| C | 61.2 | 61.5 | 65.6 | 71.2 | 83.6 | 94.0 | 98.5 | 100.0 | | |
| E | 4.6 | 5.6 | 4.1 | 2.4 | 4.6 | 2.3 | 0.6 | 0.0 | | |
| H | 34.2 | 32.9 | 30.2 | 26.4 | 11.8 | 3.7 | 0.9 | 0.0 | | |

## C.2 BASELINE DETAILS

We test three self-supervised time-lagged MLCV baselines: DeepTICA (Bonati et al., 2021), TAE (Wehmeyer & Noé, 2018), and VDE (Hernández et al., 2018). Since simulation configurations vary in each paper, we retrain all models with the same data using the `mlcolvar` library (Bonati et al., 2023). All models have the following identical configurations. We use a neural network size of [45, 100, 100, 1] with tanh as the activation function, and a dropout of 0.5. We split the dataset into 80% for training and 20% for validation. Models, including ours, were trained for a maximum of 1000 epochs, with early stopping applied with a minimum delta of 0.1 and patience of 50 epochs. Unless mentioned, we follow the basic configuration of the `mlcolvar` (Bonati et al., 2023).

## C.3 SIMPLE BASELINES

**Experiment setup.** Additionally, we show that simple baselines, e.g., PCA and TICA, on $C_\alpha$-wise distance fail for Trp-cage. We use the first principal component and the first time-lagged principal component as CVs. All other details are identical; the training data is for the baseline and BIOEMU-CV, normalization to the range [-1, 1] over the whole DESRES trajectory data.

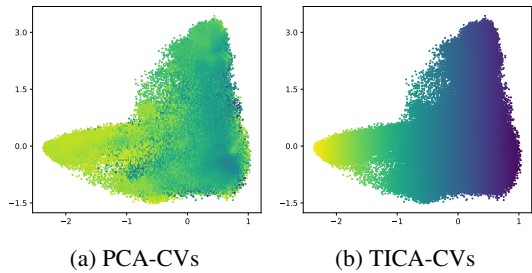

(a) PCA-CVs      (b) TICA-CVs

Figure 11: **Simple method CVs on TICA projections of the full DESRES trajectory.** PCA-CVs fails to discriminate the folded and unfolded state, while TICA-CVs obviously shows correlation with the TICA using the full DESRES trajectory.

**Qualitative results.** In Figure 11, we color TICA plot with the CVs. Since the axes were computed with TICA on the full DESRES dataset, TICA-CVs show high correlation with the $x$ axis. Nonetheless, PCA-CVs show meaningless values in Trp-cage compared to MLCVs in Figure 6, failing to discriminate between the folded and unfolded state.

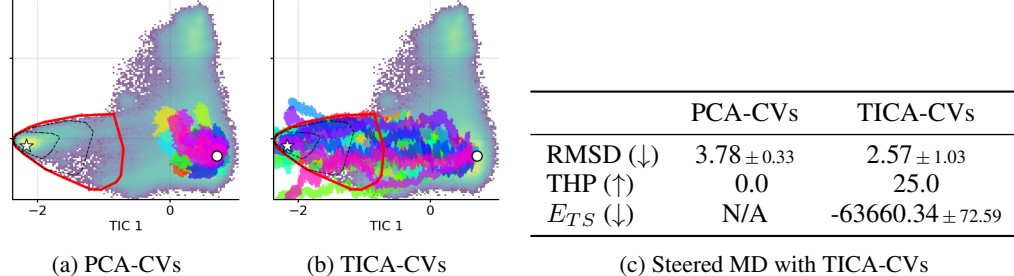

|  | PCA-CVs | TICA-CVs |
|---|---|---|
| RMSD (↓) | $3.78_{\pm 0.33}$ | $2.57_{\pm 1.03}$ |
| THP (↑) | 0.0 | 25.0 |
| $E_{TS}$ (↓) | N/A | $-63660.34_{\pm 72.59}$ |

(a) PCA-CVs    (b) TICA-CVs    (c) Steered MD with TICA-CVs

Figure 12: **Visualization of transition paths and quantitative results of steered MD.** PCA-CVs fails to reach the target state, where TICA-CVs sometimes reach the target state with a low energy.

**Steered MD results.** For steered MD, we identically mix with the $C_\alpha$ RMSD CVs in a ratio of 5:5, as for our baselines and BIOEMU-CV. In Figure 12, PCA-CVs fail to reach the target state, while TICA-CVs often reach the target state with low energy. However, its THP is low compared to baselines, failing to find meaningful transition pathways.

## C.4 OPES SIMULATION DETAILS

**Simulation settings.** OPES simulations for all proteins were performed with GROMACS 2024.3 (Abraham et al., 2015) patched with PLUMED 2.10 (Tribello et al., 2014), on top of Docker containers. Proteins were solvated in a cubic water with the same size as the reference simulation data, and each neutralized using sodium or chloride ions. The protein molecule was parameterized by the CHARMM27 force field (Piana et al., 2011), i.e., CHARMM22 plus CMAP backbone correction, and the modified TIP3P water model compatible with the CHARMM force field (MacKerell Jr et al., 1998), analogously to the reference simulation data (Lindorff-Larsen et al., 2011). Simulations were all performed in the NVT ensemble. Initial states were selected from folded states among the original DESRES trajectory, where the folded state was identified with native contacts and secondary structures. Afterward, the state was equilibrated through short NVT and NPT simulations, each of length 50 and 500 picoseconds, respectively. Temperatures were controlled with the velocity rescaling thermostat (v-rescale) (Bussi et al., 2007). Equations of motion were integrated with a time step of 2 femtoseconds with the leap-frog algorithm (Hockney et al., 1974). Additionally, the LINCS algorithm (Hess et al., 1997) was used to constrain all bonds involving the hydrogen bonds. All simulations were run on a single GPU of RTX 3090 or RTX 4090, with four to nine days depending on the protein size. Approximately 8,000 GPU hours were used for a single evaluation.

**OPES configurations.** We use configurations as in Table 12 for PLUMED, differing only in temperatures. A PACE of 500 steps is applied, i.e., 1 picoseconds. We use the same SIGMA value 0.05 and BARRIER value 30 for all systems. Temperatures are identical to the reference simulation. The first 100 ns are discarded as equilibration time with 50 ns window unit time steps for Figures 2 and 14.

Table 12: **OPES Simulation details** of three DESRES fast-folding proteins.

| Protein | PACE | SIGMA | BARRIER (kJ/mol) | Temperature (K) |
|---|---|---|---|---|
| Chignolin | 500 | 0.05 | 30 | 340 |
| Trp-cage (TC10b) | 500 | 0.05 | 30 | 290 |
| BBA | 500 | 0.05 | 30 | 325 |

**Evaluation metrics.** We evaluate the learned collective variables (CVs) using two complementary metrics from Yang et al. (2024): the free energy difference $\Delta F$ and the mean absolute error (MAE) of the potential of mean force (PMF). The free energy difference $\Delta F$ quantifies the stability gap between the folded and unfolded states. It is computed by integrating the PMF $A(s)$ for CVs $s$ over the corresponding metastable basins,

$$\Delta F = -k_B T \log \left( \frac{\int_{\text{folded}} \exp(-A(s)/k_B T) \, ds}{\int_{\text{unfolded}} \exp(-A(s)/k_B T) \, ds} \right), \tag{2}$$

where $k_B$ is the Boltzmann constant and $T$ is the temperature. A lower $\Delta F$ error indicates that the CV preserves the free energy difference between metastable states more accurately. For the folded and unfolded basins, we divided the reference CVs range to half and use them for each one.

In addition, we use the mean absolute error (MAE) of the PMF to evaluate the agreement between the biased and reference free-energy landscapes. PMF, The MAE is defined as

$$\text{MAE}(A, A_{\text{ref}}) = \frac{\int |A(s) - A_{\text{ref}}(s)| \, \mathbb{I}[A_{\text{ref}}(s) < A_{\text{thres}}] \, ds}{\int \mathbb{I}[A_{\text{ref}}(s) < A_{\text{thres}}] \, ds}, \tag{3}$$

where $A(s)$ and $A_{\text{ref}}(s)$ are the PMFs from enhanced sampling and a long unbiased trajectory, respectively, and $\mathbb{I}[\cdot]$ is an indicator function restricting the comparison to regions with reference free energy below a threshold $A_{\text{thres}} = 25$ kJ/mol. This metric captures deviations in both metastable basins and transition regions.

**Outlier simulation exclusion.** As discussed in Yang et al. (2024), enhanced sampling simulations driven by CVs may often get trapped in a local minima, leading to outlier simulation results. To address this issue, we follow the standard practice of computing the metrics after removing a single outlier run. Specifically, among the four simulations, we identify the outlier based on the final free energy difference, i.e., the run whose value deviates the most from the mean, and exclude it. Afterwards, the metrics are re-computed using the remaining three runs.

## C.5 LONGER SIMULATION RESULTS

Here, we report the results of longer OPES simulations. To be specific, we report 9829 ns $\approx$ 9.8 $\mu$s length simulation for Trp-cage using DeepTICA, computed on four RTX 3090 GPUs for one month.

Table 13: **Quantitative results of long OPES simulation on Trp-cage with DeepTICA MLCVs.**

| Time horizon | $\Delta F_{ref}$ | $\Delta F$ | $|\Delta F_{ref} - \Delta F|$ ($\downarrow$) | PMF MAE ($\downarrow$) |
|---|---|---|---|---|
| 1 $\mu$s | 3.70 | $6.53_{\pm 7.31}$ | 2.73 | $8.94 \pm 7.43$ |
| 9.8 $\mu$s | | $-3.36_{\pm 2.76}$ | 7.60 | $7.64 \pm 3.81$ |

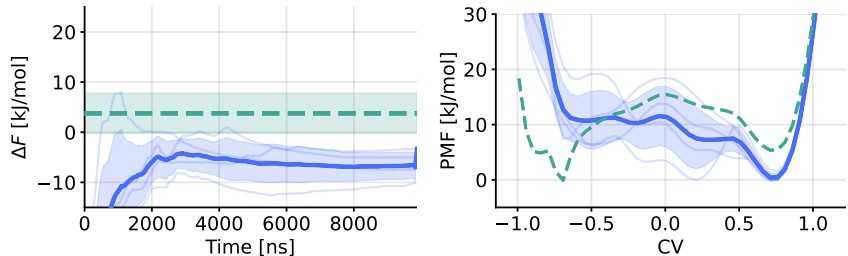

Figure 13: **Free energy (left) and PMF (right) estimation from 9.8 $\mu$ s OPES simulations for Trp-cage with DeepTICA MLCVs.** Green dotted lines indicated the reference value, and blue lines refer to the free energy difference during the OPES simulations. Solid line refer to the mean, and shaded areas are standard deviation.

As seen in Table 13 and Figure 13, simulations with approximately ten times longer results do not fully converge and still exhibit uncertainty. While the folded state is sampled many times, the unfolded states are not sampled properly, resulting in a gap in the negative range of CVs in the PMF plot. Overall, longer OPES evaluations do not provide more meaningful insights.

## C.6 STEERED MD SIMULATION DETAILS

Steered MD (Izrailev et al., 1999; Fiorin et al., 2013b, SMD) is an enhanced sampling method for sampling transition paths. It drives the state toward the target meta-stable state along the time-dependent reference CV using an additional harmonic potential. This biasing potential of SMD is defined as

$$U(x,t) = \frac{k}{2}\left\|c(x) - c_t^{\text{ref}}\right\|^2, \quad c_t^{\text{ref}} = c^{\text{initial}} + \frac{(c^{\text{target}} - c^{\text{initial}})t}{T}, \quad (4)$$

where $k$ is the force constant, $x$ the molecular configuration, $t$ the current step, $T$ the total number of steps, and $c^{\text{initial}}, c^{\text{target}}$ are the CV values at the respective meta-stable states. The reference CV starts at the initial CV $c^{\text{initial}}$ and ends at the target CV $c^{\text{target}}$ with constant rate $(c^{\text{target}} - c^{\text{initial}})/T$. The bias potential $U(x,t)$ restrains the current state to follow the reference states. To hit the target meta-stable states better, in our experiments, we combine the MLCVs $f_\theta(x)$ with (Kabsch aligned) $C_\alpha$-RMSD CV as $c(x) = f_\theta(x) - \text{RMSD}(x, x_{\text{target}})$, following the technique supported in Fiorin et al. (2013a).

We perform SMD simulations with the OpenMM software package (Eastman et al., 2023) to generate transition pathways. We use the CHARMM36 force field (Best et al., 2012) for the fast-folding proteins and the modified TIP3P model (Jorgensen et al., 1983) for water molecules. We handle long-range electrostatics via the Particle Mesh Ewald (PME) method (Ewald, 1921) with a 0.95 nm cutoff. Constraining all bonds involving hydrogen atoms allow for a 1 fs integration timestep. We run the simulations in the canonical (NVT) ensemble at 340 K. A Langevin integrator with a 1 ps$^{-1}$ friction coefficient is used for the 500 ps NVT equilibration. Finally, since the CV depends only on the $C_\alpha$ atom coordinates, we apply the biasing force, $-\nabla_x U(x,t)$, exclusively to them through the OpenMM external force class.

# D ADDITIONAL RESULTS

## D.1 OPES SIMULATION RESULTS

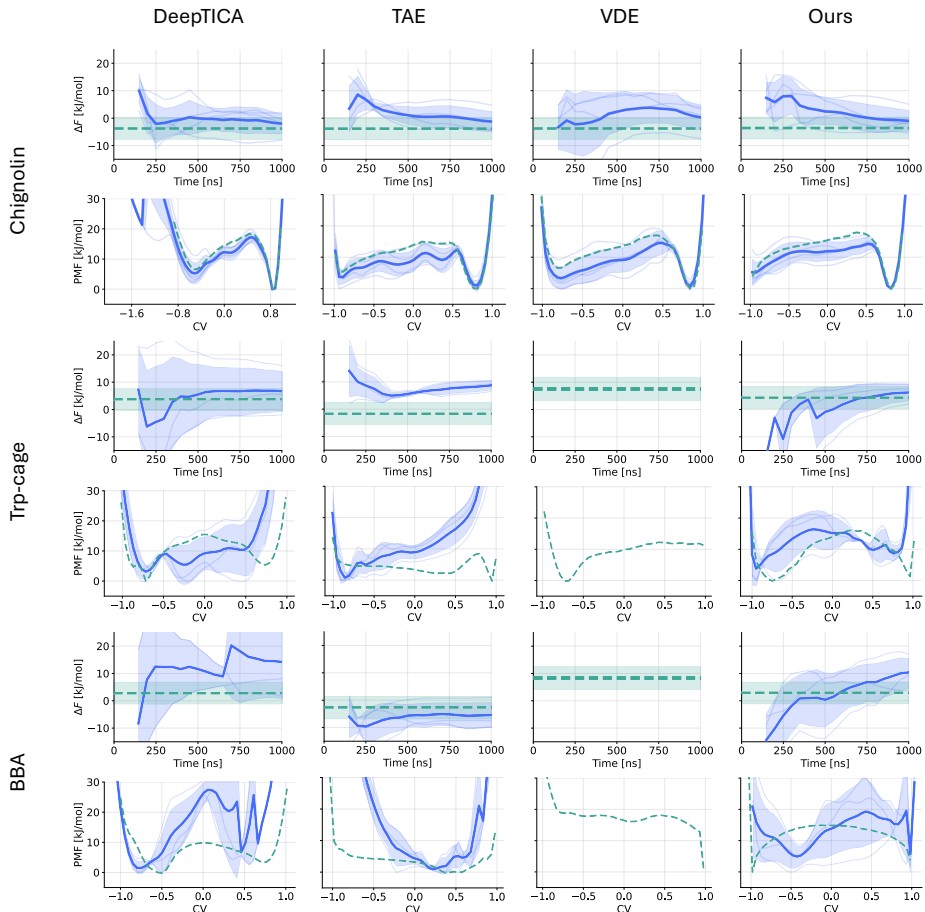

Figure 14: **Free energy difference (top) and PMF (bottom) for three proteins** from 1 $\mu s$ OPES simulation. Green dotted lines indicate the reference value computed by CVs along the full DESRES trajectory, while blue lines refer to values computed by CVs from the OPES simulations.

In Figure 14, Chignolin fairly converges while Trp-cage and BBA show relatively big deviations. In the following, we qualitatively evaluate MLCVs for each protein averaged over three simulations.

**Trp-cage.** Although TAE converged with a small deviation, this is due to failing to sample enough folded states. Also, the PMF for the folded state does not align with the reference, especially regarding the folded state region, i.e., the positive range of CVs. This indicates the failure of sampling folded states, which leads to incorrect energy value convergence. Additionally, DeepTICA shows a very low PMF compared to the reference PMF near $CV \approx 0$. This indicates that while the unfolded states were sampled properly, the folded states were not sampled enough or CVs identified as the folded states were actually transition states, resulting in a severe mismatch between PMFs.

**BBA.** Once again, although TAE appears to converge closely to the reference value, we can identify that only the folded states are exclusively sampled from the PMF plot. Unfolded states, i.e., CVs being close -1, are not sampled at all, resulting in a convergence to an incorrect value. Furthermore, the reference PMF shows only one distinct local minimum, indicating that the trained CV itself did not properly discriminate between the folded and unfolded states. While one outlier simulation has been excluded for DeepTICA, the PMF plot shows divergence between simulations near the folded states. This indicates that DeepTICA has failed to sample folded states properly, such as assigning diverse CV values to similar folded states.

## D.2 STATE DISCRIMINATION

In this section, we present the full qualitative results extended from Section 4.4. In Figure 15, we visualize the MLCV value on the z-axis on top of the TICA plane. In Figure 16, we visualize the violin plot of the MLCV of the folded and unfolded state. The states are collected from the full DESRES trajectory, with an RMSD threshold cutoff.

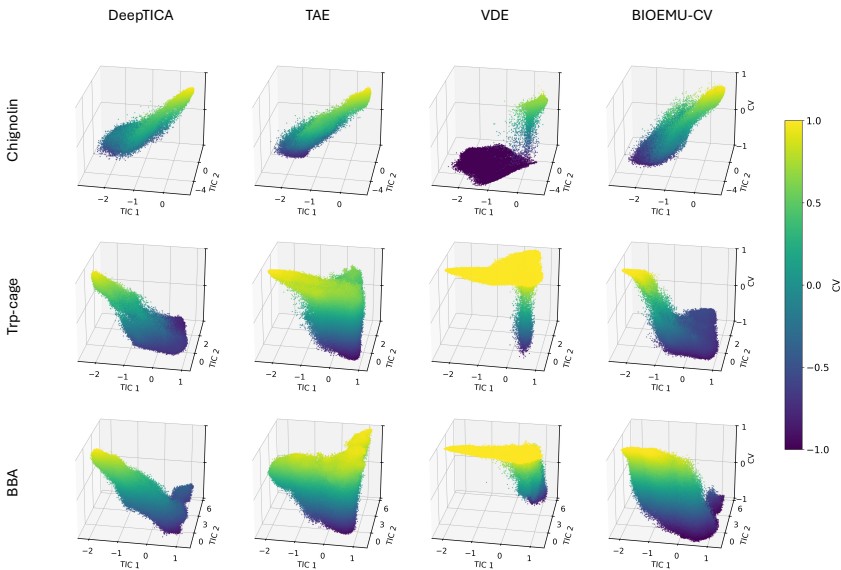

Figure 15: **3D visualization of protein conformations projected to TICA coordinates, with ML-CVs as the z axis.**

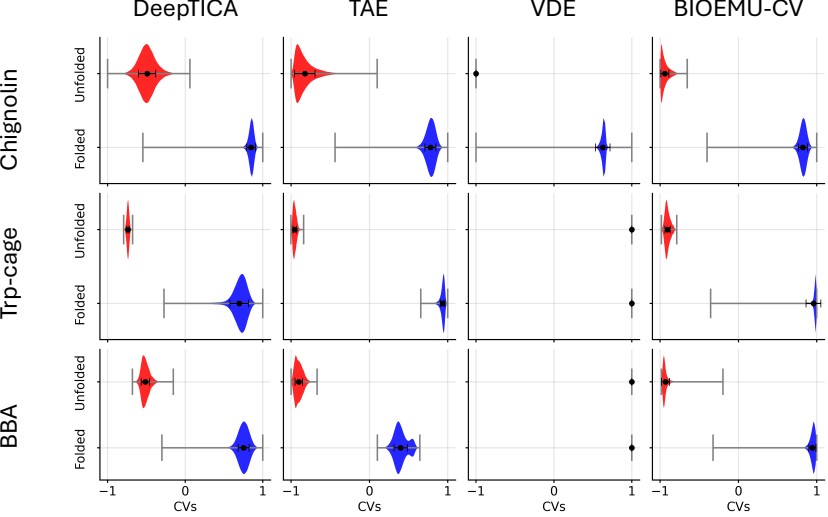

Figure 16: **MLCV distribution of the folded and unfolded states.** Average MLCV of the folded and unfolded state is stated as the black dot, with the standard deviation in black lines.

### D.3 STEERED MD VISUALIZATION

We also visualize $C_\alpha$ of MLCV-steered MD trajectory for all baselines, following Figure 3. For the folded state visualization in the DESRES data, refer to the one in Figure 6.

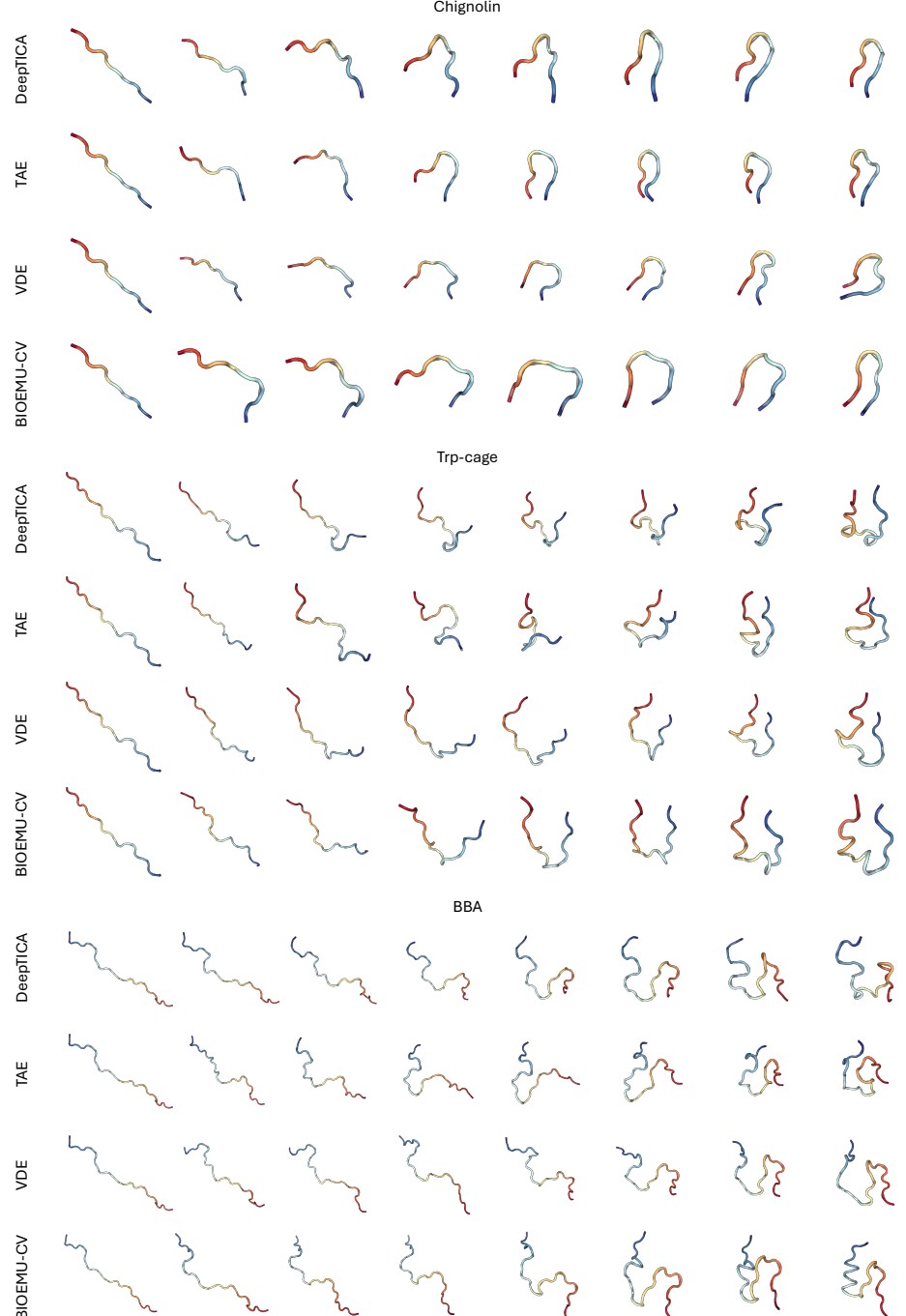

Figure 17: **3D Visualization of MLCV-steered MD paths.** The sampled folding pathways of Chignolin, Trp-cage, and BBA by steered MD with MLCVs from DeepTICA, TAE, VDE, and BIOEMU-CV.

## D.4 VAMP SCORE

Here, we evaluate the VAMP score (Noé & Nuske, 2013; McGibbon & Pande, 2015; Noé & Clementi, 2015; Wu & Noé, 2020) on MLCVs, where a higher VAMP score indicates preservation of dynamic contents. We use VAMP from `deeptime` library (Hoffmann et al., 2021), with a time-lag of 10 on the DESRES trajectory data. In Table 14, TAE and VDE show a relatively low score for bigger molecules compared to other methods. DeepTICA yields a high score since its training objective is similar to the definition of the VAMP score, and BIOEMU-CV shows similar VAMP scores in big molecules. However, VAMP scores should not be simply trusted since highly correlated values could result in high VAMP scores (Noé & Nuske, 2013; Wang et al., 2024).

Table 14: **VAMP scores of MLCVs** for Chignolin, Trp-cage, and BBA. A higher score indicates better preservation of dynamic content.

| Method | Chignolin | | | Trp-cage | | | BBA | | |
|--------|-----------|--------|--------|----------|--------|--------|--------|--------|--------|
| | VAMP-1 | VAMP-2 | VAMP-E | VAMP-1 | VAMP-2 | VAMP-E | VAMP-1 | VAMP-2 | VAMP-E |
| DeepTICA | 1.9803 | 1.9611 | 1.9611 | 1.9920 | 1.9840 | 1.9840 | 1.9898 | 1.9796 | 1.9796 |
| TAE | 1.9686 | 1.9381 | 1.9381 | 1.8787 | 1.7722 | 1.7722 | 1.8902 | 1.7925 | 1.7925 |
| VDE | 1.9850 | 1.9702 | 1.9702 | 1.9947 | 1.9894 | 1.9894 | 1.8469 | 1.7172 | 1.7172 |
| BIOEMU-CV | 1.9719 | 1.9446 | 1.9446 | 1.9939 | 1.9878 | 1.9878 | 1.9859 | 1.9721 | 1.9721 |

