# OpenReview forum: "Learning Collective Variables from BioEmu with Time-Lagged Generation"
_ICLR.cc/2026/Conference — ICLR 2026 Poster_

### Official Review · Reviewer_VbbS · 2025-10-30

**Soundness:** 3
**Presentation:** 2
**Contribution:** 2
**Rating:** 4
**Confidence:** 4

**Summary:**

This paper introduces BIOEMU-CV, a novel framework for automatically learning collective variables (CVs) for molecular dynamics (MD) simulations. The authors identify that existing machine-learning-based CVs (MLCVs) often struggle to scale from small toy systems to more complex proteins.

The proposed method cleverly leverages a large, pre-trained protein foundation model, BioEmu, as a "scaffolding" to train a new, lightweight CV encoder. The core idea is to train this encoder to find a low-dimensional CV ($c_t$) from a protein's current state ($x_t$) that is maximally predictive of a future, time-lagged state ($x_{t+\tau}$). This is achieved by freezing the weights of the powerful BioEmu model and training only the small encoder and an adapter to "condition" BioEmu to generate $x_{t+\tau}$ based on $c_t$. This time-lagged objective theoretically forces the CV to capture only the slow, persistent dynamics (e.g., folding) while discarding fast, random fluctuations.

The method is benchmarked against three other self-supervised MLCVs (DeepTICA, TAE, VDE) on three fast-folding proteins (Chignolin, Trp-cage, BBA). The evaluation is performed on two key downstream tasks: (1) free energy difference estimation using On-the-fly Probability Enhanced Sampling (OPES) and (2) transition path sampling using Steered Molecular Dynamics (SMD).

**Strengths:**

1. The paper's most significant contribution is demonstrating a clear failure of popular baseline methods (VDE and TAE) to scale beyond the 10-residue Chignolin protein. As shown in Table 3 and Figure 5, VDE is completely incapable of separating the folded and unfolded states for Trp-cage and BBA. BIOEMU-CV, by contrast, scales robustly and provides excellent state discrimination for all three proteins. This finding alone reframes the problem, suggesting that foundation-model approaches may be a necessary solution.

2. The method demonstrates superior qualitative performance. The CV provides a clear, unambiguous separation between the folded and unfolded basins (Table 3), which is the primary prerequisite for any useful CV. Furthermore, Figure 6 shows it successfully assigns distinct values to different secondary structures ($\alpha$-helix vs. $\beta$-sheet) in BBA, a task where VDE and TAE fail.

3. The SMD results (Table 2) provide the strongest quantitative evidence for the method's utility. BIOEMU-CV achieves a 93.8% target hit percentage (THP) for BBA, compared to just 18.8% for DeepTICA (the only other baseline to even run the task). This is a decisive result and shows the CV is highly effective as a reaction coordinate for guiding folding pathways.

4. The approach of re-purposing a large, frozen foundation model is computationally appealing. It avoids the cost of training a massive new model from scratch and instead produces a small, fast, and practical CV encoder that can be easily plugged into existing MD simulation packages.

**Weaknesses:**

1. This is the most significant weakness in the paper's evaluation: The OPES simulations (Table 1) are demonstrably not converged. The reported standard deviations for the free energy difference ($\Delta F$) are often larger than the mean values themselves (e.g., $1.36 \pm 7.92$ kJ/mol for Trp-cage and $0.82 \pm 16.57$ kJ/mol for BBA). With such massive error bars, it is statistically impossible to compare the methods. The authors' claims of "winning" based on the mean value are unsupportable. These results should be treated as inconclusive.

2. Even within the inconclusive OPES results, the "win" is not uniform. For Chignolin (the most well-behaved system), DeepTICA achieves a notably better PMF-MAE (2.64) than BIOEMU-CV (3.07).

3. The target state $x_{t+\tau}$ is a synthetic construct (BioEmu C$\alpha$ + hpacker + relaxation), not a pure MD frame. This introduces a significant potential for bias, where the CV may be learning the artifacts of the hpacker algorithm rather than the true, all-atom slow dynamics. This concern is not addressed with an ablation study (this point is expanded in Question 1).

4. The method is benchmarked on three "fast-folding proteins". While this is an improvement over systems like alanine dipeptide, these are still relatively small proteins (10-28 residues). Its scalability to much larger, more complex proteins (e.g., 100+ residues) is not demonstrated.

5. The study is limited to a 1-dimensional CV ($d=1$) for simplicity. While this is readable, it is likely limiting for multi-pathway kinetics or larger proteins; the paper doesn’t probe $d>1$ (this point is expanded in Question 3).

**Questions:**

1. The target state $x_{t+\tau}$ is not a pure simulation frame but a synthetic construct (BioEmu C$\alpha$ + hpacker side chains + OpenMM relaxation). This pipeline may introduce its own biases. How can the authors be sure that the learned CV isn't just learning to predict the artifacts of the hpacker algorithm rather than the true underlying slow dynamics of the protein? An ablation study training on "native" MD frames vs. the synthetic targets would seem necessary to resolve this.

2. Given the massive standard deviations in Table 1, the 1µs OPES simulations are clearly unconverged. Can the authors provide longer-run data or, failing that, justify why any conclusions are being drawn from this specific task? As it stands, the $\Delta F$ results are statistically inconclusive and weaken the paper's overall argument.

3. The study is limited to a 1-dimensional CV ($d=1$) for simplicity. However, complex folding events often require more than one coordinate. How does the method perform when tasked with finding 2 or 3 CVs? Does the time-lagged generation approach successfully separate multiple slow modes?

4. The steering variable for the SMD task is a combination of the learned CV and C$\alpha$-RMSD. This is a significant confounding variable. Can the authors please provide results for the SMD task using only the learned CV as the steering variable? This is necessary to prove that BIOEMU-CV itself, and not the C$\alpha$-RMSD helper, is responsible for the high target hit percentage.

5. The paper does not include "simple baseline" controls. How does the learned CV's performance (e.g., in SMD THP) compare to a simple, non-learned CV, such as a linear PCA or TICA component derived from either the input coordinates or the internal latents of the frozen BioEmu model? This is needed to demonstrate that the complex, time-lagged training is necessary.

---

> ### Author Response · Authors · 2025-11-25
>
> **[W1, W2, Q2] Large deviation in OPES simulation makes results inconclusive.**
>
> We thank the reviewer for bringing this important point to our attention. We acknowledge that OPES simulations exhibit large deviations, also empirically observed in prior works [1-3]. To address these concerns, we (a) explain why such deviations are inherent, (b) provide updated results with smaller deviations, and (c) clarify why our experiments remain meaningful despite the deviations.
>
> **Deviation inherent in OPES simulations.**
> Independent OPES simulations naturally show variability due to the stochasticity in molecular dynamics and rare-event sampling, as reported in prior works [1-3]. Furthermore, we demonstrate that longer OPES simulations do not necessarily resolve this issue. We provide a 9.8 $\mu$s run OPES simulation for DeepTICA on Trp-cage in Appendix C, where deviations persist despite the extended simulation time. This supports our view that relying on a single metric, e.g., the difference in $\Delta F$ to a reference, is insufficient and motivates our use of multiple results and analyses.
>
> **Updated results with smaller deviation.**
> To alleviate the reviewer's concern, we follow the established protocols to systematically exclude outlier simulations [1,4,5] for Trp-cage and BBA. To be specific, we exclude a single outlier simulation among four independent simulations, based on the deviation of its $\Delta F$ from the mean. We have updated Table 1 using this procedure and added details in Appendix B (note that DeepTICA on BBA remains unchanged, as one run was already excluded due to NaN values).
>
> **Role and interpretation of OPES simulation results.**
> We do not claim that OPES alone provides a fully conclusive ranking of methods; rather, we emphasize that our conclusion stems from broader evaluations, e.g., transition path sampling and diverse qualitative analyses. Across these evaluations, BioEmu-CV performs at least comparably to or outperforms existing MLCVs. For example:
> - TAE on Trp-cage (OPES): While $\Delta F$ appears converged, the PMF (Figure 10) shows that the simulation predominantly samples unfolded states, leading to an apparently stable but incorrect value with the wrong sign.
> - TAE on BBA (OPES): The reference PMF fails to discriminate between the folded and unfolded states, due to inadequate state definitions.
> - DeepTICA on BBA (SMD): DeepTICA yields large RMSD, high maximum energy, and low THP compared to BioEmu-CV, indicating poor steering performance.
> - VDE (state discrimination): VDE fails to properly discriminate metastable states, therefore unsuitable as CVs for large proteins.
>
> Taken quantitative and qualitative results together, even acknowledging the inherent deviations in OPES, the overall evidence supports our conclusion for the performance of BioEmu-CV.
>
>
>
> **[W3, Q1] The target state is a synthetic construct (BioEmu C + hpacker side chains + OpenMM relaxation).**
>
> As mentioned in our paper, our target state is not a synthetic construct but a pure simulation frame from the DESRES trajectory. To be specific, the model conditions on data $x_t$ and learns to generate structures matching data $x_{t+\tau}$. All experiments in this paper are performed at residue-level resolution, where hpacker and OpenMM relaxation are not used. We made this point clearer in the method section.
>
>
>
> **[W4] Experiments limited to small proteins.**
>
> We appreciate the reviewer's comment regarding protein size, but we would like to clarify that (1) the chosen proteins are comparable to or larger than those typically examined in current literature on CVs for proteins [1, 6-8] and (2) evaluations are limited to small proteins mainly due to the reliable validation of collective variables. As the reviewer has mentioned, our work is an improvement over small systems such as Alanine Dipeptide, and to our knowledge, it is the first study to provide a systematic comparison of diverse MLCVs both qualitatively and quantitatively.
>
> While BioEM itself is designed to handle larger proteins, the practical limitation is the computation cost of OPES-based evaluation. For instance, a single MLCV evaluation on BBA took nine days with four RTX 3090 GPUs, and a 100-residue protein would require more than a month. Given these constraints, we believe that the standard and rigorous validation on well-characterized systems is essential before addressing larger proteins.

---

> ### Author Response · Authors · 2025-11-25
>
> **[W5, Q3] Experiments being limited to 1-dim CVs and multi-pathway kinetics.**
>
> To resolve the reviewer's concern, we present additional experimental results of multi-dimensional CVs and testing CVs on a multi-path system.
>
> **Multi-dimensional CVs**
>
> We extend our framework to 4-dimensional CVs on BBA and present qualitative results in Appendix C. Since obtaining quantitative metrics for multi-dimensional CVs is non-trivial, we instead (i) perform sensitivity analysis to show that individual CV dimensions respond to different meaningful slow degrees of freedom, and (b) measure their correlation with the secondary structures formation during folding trajectories. In Appendix C, these analysts confirm that our framework scales effectively to higher-dimensional CVs.
>
>
> **Multi-pathway kinetics**
>
> We additionally report the results of 1-dim CVs steered MD on Alanine Dipeptide, a system known to show multi-pathway kinetics [9, 10]. To be specific, two transition pathways appear from $C5$ to $C7_{ax}$ crossing saddle points near $\phi \approx 0$. We use 1-dimensional CVs learned with the Transferable Boltzmann Generator backbone [11] at heavy-atom resolution (BioEmu-CV is not applicable here due to its residue-level input). We report the full results in Appendix C.
>
> | Method | RMSD ($\downarrow$) | THP ($\uparrow$) | Energy ($\downarrow$) |
> | - | - | - | - |
> | DeepTICA | 0.9729 | 8.59 | 814.52 $\pm$ 115.74 |
> | TAE | 1.0086 | 58.59 | 755.41 $\pm$ 92.30 |
> | VDE | 0.8582 | 5.08 | 901.69 $\pm$ 115.59 |
> | Ours | 0.9593 | 60.93 | 33.58 $\pm$ 15.19 |
>
> Quantitative metrics and transition path visualization show that our 1-dimensional CVs successfully capture two transition pathways traversing near the saddle points, while baselines either fail to reach the target or collapse to a single pathway. This suggests that our framework can capture multi-path kinetics and is promising to extend to higher-dimensional CVs.
>
>
>
>
> **[Q4] SMD without mixing learned CV with RMSD CV**
>
> We provide steered MD results using only MLCVs, with the force constant $k$ chosen to avoid abnormally high energies. We have added the full results in Appendix D of the updated manuscript.
>
> Chignolin ($k=1000$)
> | MLCV | RMSD ($\downarrow$) | THP ($\uparrow$) | Energy ($\downarrow$) |
> |-|-|-|-|
> | DeepTICA | 7.24 $\pm$ 1.49 | 0.0 | N/A |
> | TAE | 5.72 $\pm$ 1.94 | 6.2 | -81951.53 $\pm$ 0 |
> | VDE | N/A | N/A | N/A |
> | BioEmu-CV | 7.82 $\pm$ 1.47 | 0.0 | N/A |
>
> Trp-cage ($k=5000$)
> | MLCV | RMSD ($\downarrow$) | THP ($\uparrow$) | Energy ($\downarrow$) |
> |-|-|-|-|
> | DeepTICA | 10.78 $\pm$ 1.15 | 0.0 | N/A |
> | TAE | 7.22 $\pm$ 1.71 | 0.0 | N/A |
> | VDE | N/A | N/A | N/A |
> | BioEmu-CV | 7.14 $\pm$ 1.52 | 0.0 | N/A |
>
> BBA ($k=50000$)
> | MLCV | RMSD ($\downarrow$) | THP ($\uparrow$) | Energy ($\downarrow$) |
> |-|-|-|-|
> | DeepTICA | 7.39 $\pm$ 2.71 | 0.0 | N/A |
> | TAE | 9.87 $\pm$ 1.59 | 0.0 | N/A |
> | VDE | N/A | N/A | N/A |
> | BioEmu-CV | 6.03 $\pm$ 2.56 | 0.0 | N/A |
>
> We mark N/A for simulation results that do not reach the target or diverge with numerical errors. Across all systems, steered MD with MLCVs alone fails to reach the target state. For this reason, we follow prior work [12] and mix learned CVs with RMSD-CVs. As one can see in Figure 4 of the original paper, prior MLCVs mixed with RMSD-CVs still mostly fail to reach the target state, while BioEmu-CV reaches the target state with a relatively high probability and lower energy.

---

> ### Author Response · Authors · 2025-11-25
>
> **[Q5] Simple baseline comparison (PCA or TICA on input coordinates or internal latent) for steered MD.**
>
> **PCA and TICA on input coordinates**
>
> To resolve the reviewer's concern, we have tested simple baselines for steered MD: the first principal component from PCA and the first time-lagged independent component from TICA as CVs, given $C_\alpha$-wise distances as inputs. We have mixed RMSD-CVs as in Table 2, and present the full result in Appendix C.
>
> Trp-cage ($k=20000$)
> | MLCV | RMSD ($\downarrow$) | THP ($\uparrow$) | Energy ($\downarrow$) |
> |-|-|-|-|
> | PCA | 3.78 $\pm$ 0.33 | 0.0 | N/A |
> | TICA | 2.57 $\pm$ 1.03 | 25.0 | -63660.34 $\pm$ 72.59 |
> | DeepTICA | 2.37 $\pm$ 0.47 | 31.2 | -63611.88 $\pm$ 57.49 |
> | TAE | 2.75 $\pm$ 0.35 | 0.0 | N/A |
> | VDE | N/A | N/A | N/A |
> | BioEmu-CV | 2.31 $\pm$ 0.52 | 31.2 | -63787.51 $\pm$ 31.23 |
>
> Quantitative and qualitative results show that both PCA and TICA struggle to reach the target state, showing higher RMSD and lower THP.
>
> **PCA and TICA on internal latents**
>
> Constructing PCA and TICA baselines on BioEmu's internal latents is non-trivial in our setting. BioEmu is a denoising diffusion model only conditioned on the sequence; therefore, there is no straightforward way to map coordinates back into its latent space. Designing such a pipeline would require substantial additional modelling and engineering, and we regard it as an interesting but separate direction for future work.
>
>
>
> [1] Yang et al., "Learning Collective Variables with Synthetic Data Augmentation through Physics-Inspired Geodesic Interpolation", Journal of Chemical Theory and Computation 2024
>
> [2] Rizzi et al., "The Arch from the Stones: Understanding Protein Folding Energy Landscapes via Bioinspired Collective Variables", The Journal of Chemical Physics Letter 2025
>
> [3] Marinelli, et al. "A kinetic model of trp-cage folding from multiple biased molecular dynamics simulations.", PLoS computational biology 2009
>
> [4] Jing, et al. "A highly accurate metadynamics-based Dissociation Free Energy method to calculate protein–protein and protein–ligand binding potencies." Scientific Reports 2022
>
> [5] Schulze et al., "All you need is water: Converging ligand binding simulations with hydration collective variables", The Journal of Chemical Physics 2025
>
> [6] Wu et al., "Calculating linear and nonlinear multi-ensemble slow collective variables for protein folding", The Journal of Chemical Physics 2024
>
> [7] Faran et al., "A Stochastic Landscape Approach for Protein Folding State Classification", Journal of Chemical Theory and Computation 2024
>
> [8] Rizzi et al., "The Arch from the Stones: Understanding Protein Folding Energy Landscapes via Bioinspired Collective Variables", The Journal of Chemical Physics Letter 2025
>
> [9] Holdijk et al., "Stochastic Optimal Control for Collective Variable Free Sampling of Molecular Transition Path", NeurIPS 2023
>
> [10] Seong et al., "Transition Path Sampling with Improved Off-Policy Training of Diffusion Path Samplers", ICLR 2025
>
> [11] Klein et al., "Transferable Boltzmann Generators." NeurIPS 2024
>
> [12] Fiorin et al., "Using collective variables to drive molecular dynamics simulations", Molecular Physics 2013

---

> > ### Comment · Reviewer_VbbS · 2025-11-26
> >
> > I thank the authors for their detailed and constructive rebuttal. I will increase my score.
> >
> > The authors have effectively resolved my primary methodological concern regarding the target state construction. The clarification that training occurs at residue-level resolution on real DESRES frames (bypassing hpacker artifacts) removes the potential for bias I initially feared. Furthermore, the new SMD ablations are convincing; showing that BioEmu-CV + RMSD drastically outperforms DeepTICA + RMSD (while simple baselines fail) proves the learned CV provides essential guidance for the folding process.
> >
> > Remaining Reservations:
> > While I appreciate the context regarding OPES variance, the fact remains that the primary quantitative metric in the paper ($\Delta F$ in Table 1) is statistically noisy. Even with the exclusion of outliers, the error bars are substantial, making it difficult to claim a definitive quantitative improvement in free energy estimation over converged baselines.
> >
> > While the method solves the scaling failure of VDE/TAE (a strong contribution), the validation is still limited to relatively small fast-folding proteins. Given the use of a "foundation model," the demonstration of scalability to larger, complex systems remains a theoretical promise rather than an empirical result in this work.

---

> > > ### Author Response · Authors · 2025-11-30
> > >
> > > We thank the reviewer for the follow-up and positive reassessment.
> > >
> > > Additionally, we respectfully clarify that the OPES evaluation serves primarily to assess sampling feasibility under challenging conditions, rather than to provide a precise measure of convergence. The key result is that our method successfully navigates transitions where baselines fail. For instance, TAE predicts the wrong sign because it samples mostly unfolded states in Trp-cage, failing to drive transitions.
> > >
> > > We sincerely thank the reviewer for their time and valuable insights throughout the process.

---

### Official Review · Reviewer_UdyY · 2025-10-31

**Soundness:** 3
**Presentation:** 3
**Contribution:** 3
**Rating:** 8
**Confidence:** 3

**Summary:**

Enhanced sampling uses collective variables (CVs) for sampling from long timescales which can represent rare events. However, finding good CVs is not trivial. This paper proposes to learn these from BioEmu. Concretely, an encoder is trained to embed the input structure to a low-dimensional vector, which represents the CVs. This vector is added to the single representation in BioEmu so that the time-lagged conformation is generated.

To evaluate the model, it looks at the free energy difference estimation and transition path sampling, which requires correct long timescale encoding. Comparisons to the baselines show comparable/better performance of the proposed model.

**Strengths:**

* originality
 Learning latent representations from a fixed pretrained model is not new but this paper adapts it for learning CVs for slow dynamics which is a novel application.
 * quality
 The model's predictions are compared against state-of-the-arts quantitatively while it would have been better if more that 3 proteins were tested.
 * clarity
 It is mostly clear to understand except for some typos and sentences.
 * significance
 Running MD simulations to investigate slow dynamics of proteins, e.g. folding, is challenging due to the required computation resource and error accumulation. Enhanced sampling methods can tackle this challenge but it needs good VCs. This paper introduces an effective way of learning them by wrapping around an existing protein ensemble generation model, BioEmu. The results are competitive and extensive comparisons were conducted. The folding path sampling results show a good performance of the model.

**Weaknesses:**

* To see if the model works well on a variety of proteins, it would make the paper stronger if more than 3 proteins were tested.

**Questions:**

* There are two downstream tasks used: free energy difference estimation and transition path sampling. What could be other metrics to used to measure the effective encoding of slow dynamics?
* The proposed model is compared against self-supervised models. Are there any supervised models for the task?
* Line309: 'not meaningless'->'meaningless'?
* line317: 'through out'->'throughout'
* Fig1 caption: 'earn'->'learn'
* Fig4, the circle represents the folded state. Why is the red convex hull not around it for chignolin, while it is for others? Also, would it be better to draw lines for transition paths somehow, to clearly see the paths?
* Fig3, it would be interesting to also show the transitions of samples from compared models.
* Have you tried using other models than BioEmu, e.g. Boltz, AlphaFlow? I guess it should be straightforward to do it and should be interesting to compare the results.

---

> ### Author Response · Authors · 2025-11-25
>
> We thank the reviewer for going over the paper carefully, as well as for suggesting several improvements. We have incorporated the reviewer's feedback on the updated manuscript, marked in purple.
>
> **[W1] Experiments being limited to a small number of proteins.**
>
> We agree that evaluating the suggestion of testing our method on a broader set of proteins would strengthen our work. We also emphasize, however, that our experiment covers most proteins that CVs are studied in the current literature [1-3]. In addition, we applied our framework to Alanine Dipeptide at heavy atom resolution and report the results in Appendix C of the updated manuscript. Our CVs demonstrate superior performance in steered MD, while prior methods sometimes fail to reach the target.
>
>
> **[Q1] Additional metrics to measure the encoding of slow dynamics.**
>
> To alleviate your concern, we additionally report the VAMP-2 score of MLCVs. While VAMP scores are used for evaluating dynamical information, prior works [4,5] emphasize that they should be interpreted with caution since simple correlations in the data can artifically inflate the score.
>
> VAMP-2 score
> | Method | Chignolin | Trp-cage | BBA |
> | - | - | - | - |
> | DeepTICA | 1.9611 | 1.9840 | 1.9796 |
> | TAE | 1.9381 | 1.7722 | 1.7925 |
> | VDE | 1.9702 | 1.9894 | 1.7172 |
> | BioEmu-CV | 1.9446 | 1.9878 | 1.9721 |
>
> BioEmu-CV results achieve high VAMP-2 scores in large proteins (Trp-cage and BBA), indicating that it captures meaningful dynamical modes. Note that, while DeepTICA achieves a high VAMP-2 score due to its similarity between its training objective and the VAMP score formulation, it underperforms on committor-based analysis in Table 4 and Figure 8 of the original paper. We further report all VAMP scores [4] in Appendix C.
>
>
> **[Q2] Existence of related works on supervised CV learning.**
>
> As we have mentioned in Section 2 of our original manuscript, DeepLDA and DeepTDA are supervised methods for learning CVs. However, supervised approaches require predefined binary state labels, which in practice depend on robust clustering from long equilibrium simulations. Therefore, we focus on self-supervised methods in this study. We clarified this point more in the updated manuscript.
>
>
> **[Q3-7] Typos, figure modification, and additional visualization of steered MD.**
>
> We thank the reviewer for pointing out minor typos and improving the paper! We have updated the manuscript accordingly. For Figure 4, we corrected the placement of the circle for Chignolin. Additionally, following Figure 3, we added a visualization of every MLCV-steered MD in Appendix C, as requested.
>
>
> **[Q8] Other generative models as backbone.**
>
> We have chosen BioEmu for its ability to generate diverse conformations proportional to energy; however, AlphaFlow could also be applied, as the reviewer mentioned. To demonstrate this, we applied our framework to the Transferable Boltzmann Generators backbone [6] on the Alanine Dipeptide, as BioEmu is not applicable due to its residue-level inputs. The results are provided in Appendix C and show that our approach also outperforms baselines in steered MD.
>
>
> [1] Wu et al., "Calculating linear and nonlinear multi-ensemble slow collective variables for protein folding", The Journal of Chemical Physics 2024
>
> [2] Faran et al., "A Stochastic Landscape Approach for Protein Folding State Classification", Journal of Chemical Theory and Computation 2024
>
> [3] Rizzi et al., "The Arch from the Stones: Understanding Protein Folding Energy Landscapes via Bioinspired Collective Variables", The Journal of Chemical Physics Letter 2025
>
> [4] N&#243;e and Feliks Nuske, "A variational approach to modeling slow processes in stochastic dynamical systems", Multiscale Modeling & Simulation 2013
>
> [5] Wang et al., "Information bottleneck approach for markov model construction", Journal of Chemical Theory and Computation 2024
>
> [6] Klein et al., "Transferable boltzmann generators." NeurIPS 2024

---

### Official Review · Reviewer_F5dg · 2025-10-31

**Soundness:** 2
**Presentation:** 2
**Contribution:** 2
**Rating:** 4
**Confidence:** 3

**Summary:**

This work proposes a novel framework for learning collective variables. The model trains an encoder to extract CVs by enabling the frozen generative model BioEMU to generate molecular conformations under time-lagged conditions. The paper benchmarked MLCVs for two slow degree freedom tasks and demonstrated the model's effectiveness.

**Strengths:**

1. The idea of capturing collective variables through a conditional generative task is novel.
2. The paper is well-structured, clearly presented, and supported by thorough experiments.

**Weaknesses:**

1. The method heavily relies on the model capabilities of BioEmu. The captured CV may contain model bias.
2. The paper a bit overstates the scope of its CVs, especially in the title. It should make clear that the method focuses on CVs for enhanced sampling, rather than general CV learning.
3. The dataset is quite limited, and the selected proteins are all very small. The method should be tested on a broader set of proteins with varying sizes to better demonstrate its generality and robustness.

**Questions:**

1. How physically interpretable are the learned CVs?
2. If BioEmu’s modeling capabilities are insufficiently accurate, will the learned CV still reflect the system's true slow degrees of freedom? How can this issue be mitigated?

---

> ### Author Response · Authors · 2025-11-25
>
> We thank the author for carefully reading our paper, and suggesting additional analysis to strengthen our approach. Specifically, we have added the sensitivity analysis as Section 4.3 in our updated manuscript (marked in purple), where BioEmu-CV shows good interpretability for the distances related to folding.
>
>
> **[W1, Q2] Heavy reliance on BioEmu.**
>
> To alleviate the concern that our method is reliant on BioEmu, we additionally test the same framework with a different backbone, the transferable Boltzmann generator architecture [1] for Alanine Dipeptide. Below, we report the performance of steered MDs.
>
> | Method | RMSD ($\downarrow$) | THP ($\uparrow$) | Energy ($\downarrow$) |
> | - | - | - | - |
> | DeepTICA | 0.9729 | 8.59 | 814.52 $\pm$ 115.74 |
> | TAE | 1.0086 | 58.59 | 755.41 $\pm$ 92.30 |
> | VDE | 0.8582 | 5.08 | 901.69 $\pm$ 115.59 |
> | Ours | 0.9593 | 60.93 | 33.58 $\pm$ 15.19 |
>
> As shown in the above table and steered MD path visualization in Appendix C, our framework shows higher performance and successfully identifies the transition path regarding the energy landscape, unlike prior works that often ignore the energy landscape. This demonstrates that our method is not tied to BioEmu specifically, but can be paired with other generative models.
>
> We additionally advocate for the use of BioEmu, since it is trained over diverse conformations from MD trajectories and generates protein conformations regarding the energy distributions, resulting in a powerful model. Nonetheless, if BioEmu's modeling capability fails in a particular system, a practical mitigation would be to run short MD simulations with negligible costs, and additionally fine-tune BioEmu.
>
>
>
> **[W2] Modify title to specify enhanced sampling.**
>
> We have modified the title as "*Learning Collective Variables for Enhanced Sampling from BioEmu with Time-Lagged Generation*" in our updated manuscript to incorporate the reviewer's concern. Nevertheless, we note that our analysis goes beyond enhanced sampling: we also study various analyses, including meta-stable state discrimination, secondary structure behavior, and sensitivity analysis.
>
>
> **[W3] Experiments being limited to small proteins.**
>
> While we understand the reviewer's concern, we respectfully note that (1) proteins used in this study are comparable in size or larger than those in the current literature [2-5], and (2) evaluations are limited to small proteins mainly due to the ability to validate the collective variables. To our knowledge, this is the first work on a systematic qualitative and quantitative comparison of MLCVs. While extending to larger proteins is a valuable direction, OPES-based evaluation is computationally demanding. For example, a single MLCV evaluation on BBA took nine days with four RTX 3090 GPUs, and a 100-residue protein would approximately take more than one month. We believe that carefully validating on well-characterized systems is a necessary prerequisite before addressing larger, unverified systems.
>
>
> **[Q1] Physical interpretability of CVs.**
>
> In Section 4.3 of the updated manuscript, we added new experimental results on the interpretability of CVs through the sensitivity analysis. We compute the sensitivity of each MLCVs to its input features, i.e., $C_\alpha$ wise distances, with the `mlcolvar` library [6]. We then visualize the most sensitive distances in both the folded and unfolded states. Surprisingly, BioEmu-CV results in high sensitivity to long-range contacts that are known to be important for folding. For example, in Chignolin, BioEmu-CV is particularly sensitive to the distance between TYP1 and TYR10, where a hydrogen bond is observed at folding [4].
>
>
> [1] Klein et al., "Transferable Boltzmann Generators." NeurIPS 2024
>
> [2] Wu et al., "Calculating linear and nonlinear multi-ensemble slow collective variables for protein folding", The Journal of Chemical Physics 2024
>
> [3] Faran et al., "A Stochastic Landscape Approach for Protein Folding State Classification", Journal of Chemical Theory and Computation 2024
>
> [4] Yang et al., "Learning Collective Variables with Synthetic Data Augmentation through Physics-Inspired Geodesic Interpolation", Journal of Chemical Theory and Computation 2024
>
> [5] Rizzi et al., "The Arch from the Stones: Understanding Protein Folding Energy Landscapes via Bioinspired Collective Variables", The Journal of Chemical Physics Letter 2025
>
> [6] Bonati, Luigi, et al. "A unified framework for machine learning collective variables for enhanced sampling simulations: mlcolvar.", The Journal of Chemical Physics 2023

---

### Official Review · Reviewer_H3Mz · 2025-11-03

**Soundness:** 3
**Presentation:** 3
**Contribution:** 2
**Rating:** 4
**Confidence:** 3

**Summary:**

The paper proposes a method for efficient molecular dynamics by learning collective variables (CV) from the BioEmu model. The paper adds a small encoder on top of a frozen BioEmu diffusion model to learn a one-dimensional collective variable (CV) using a time-lagged objective. The authors evaluate the learned CV on three fast-folding proteins using OPES (for free-energy and PMF error) and SMD (for target hitting and maximum energy along paths). All CV’s are restricted to 1 dimension.

**Strengths:**

- The integration is simple and practical: keep BioEmu frozen and train only a small conditioning head. This is an attractive engineering path if it works broadly.


- On SMD tasks, the method often reports better target-hitting probability and lower maximum energy along the path, which suggests the CV is useful for guiding transitions.


- The paper provides reasonably clear setup details for OPES and SMD, which helps with reproducibility.

**Weaknesses:**

- The paper needs more ablation. It is unclear which components matter most. For example, comparisons for time-lagged vs. non-time-lagged training, frozen BioEmu vs. partial fine-tuning, different encoder sizes and placements, and SMD runs without mixing the learned CV with RMSD. Without these tests, it is hard to attribute the reported gains to the proposed choices, and hard to evaluate the novelty of the proposed method vs. the reported baseline methods.
- Restricting the CV to one dimension and forcing a fixed range can hide multi-modal kinetics or make methods look closer or farther apart depending on alignment. While reported CV’s don’t need to be that high dimensional, understanding if the method scales to multi-dimensional CV’s at all is important for future practitioners.


- Several results have large standard deviations. In Tables 1 and 2, many method averages fall within each other’s uncertainty ranges. This weakens the ranking of methods and makes performance differences difficult to trust.

**Questions:**

- How much of the gain comes from time-lagging itself versus using BioEmu features?


- Does partially fine-tuning BioEmu help or hurt compared to freezing it?


- How sensitive are results to encoder size and to the choice of where conditioning is attached within the model?


- Does the approach generalize to larger proteins or multi-path systems, and to higher-dimensional CVs?

---

> ### Author Response · Authors · 2025-11-25
>
> We thank the reviewer for carefully reading our paper, and suggesting ablation studies to rigorously validate our framework. Below, we resolve the raised concerns one by one. All additional experiments and results are added in Appendices C and D of the updated manuscript.
>
> **[W1, Q1, Q2] Lack of ablation studies.**
>
> We appreciate the reviewer's request for additional experiments, and have conducted the requested ablation studies and verified the effectiveness of (i) time-lagged conditioning, (ii) frozen BioEmu weights, (iii) encoder size and conditioning placement, and (iv) mixing MLCVs with RMSD-CVs.
>
> **Time-lagged conditioning and frozen BioEmu weights**
>
> We compare our method with two variants that (a) conditions on non-time-lagged state $x_{t}$ instead of time-lagged state $x_{t+\tau}$ and (b) fine-tuning the BioEmu weights rather than freezing. The results below show that our current design choice achieves the best steered-MD performance for Chignolin. Update results are included in Appendix C.
>
> |  | $\Delta F_{ref}$ | $\Delta F$ | $\vert \Delta F_{ref} - \Delta_F \vert (\downarrow)$ | PMF MAE ($\downarrow$) | RMSD ($\downarrow$) | THP ($\uparrow$) | Energy ($\downarrow$) |
> |-|-|-|-|-|-|-|-|
> | BioEmu-CV | -3.71 | -3.19 $\pm$ 3.97 | 0.52 | 3.07 $\pm$ 2.53 | 1.20 $\pm$ 0.33 | 100.0 | -82055.15 $\pm$ 98.48 |
> | w.o. time-lag | -3.68 | -5.78 $\pm$ 3.20 | 2.10 | 1.41 $\pm$ 1.5 | 1.57 $\pm$ 0.36 | 81.3 | -82084.68 $\pm$ 62.86 |
> | Unfrozen BioEmu | -4.47 | -3.25 $\pm$ 0.81 | 1.22 | 3.53 $\pm$ 3.73 | 1.62 $\pm$ 0.31 | 100.0 | -82076.42 $\pm$ 98.20|
>
> Note that the performance of BioEmu-CV slightly varies from the original manuscript due to different computational environment.
>
> **Encoder size and conditioning placement**
>
> We increased the encoder size by factors of two and four, and show that the performance remains largely unchanged despite a substantial increase in the number of parameters.
>
> | Encoder params | RMSD ($\downarrow$) | THP ($\uparrow$) | Energy ($\downarrow$) |
> | - | - | - | - |
> | 48K (paper) | 1.20 $\pm$ 0.33 | 100.0 | -82055.15 $\pm$ 98.48 |
> | 196K | 1.41 $\pm$ 0.32 | 93.8 | -82049.40 $\pm$ 98.52 |
> | 1.27M | 1.50 $\pm$ 0.30 | 100.0 | -82071.13 $\pm$ 100.08 |
>
> Additionally transition path visualizations updated in Appendix D show similar trend.
>
> We also evaluated different conditioning placements for the encoder: (i) conditioning only the pair representation, and (ii) conditioning both the single and pair representation. One can observe that both alternatives degrades the performance, compared to our current design choice.
>
> | Conditioning representation | RMSD ($\downarrow$) | THP ($\uparrow$) | Energy ($\downarrow$) |
> | - | - | - | - |
> | Single (paper) | 1.20 $\pm$ 0.33 | 100.0 | -82055.15 $\pm$ 98.48 |
> | Pair | 1.79 $\pm$ 0.45 | 56.3 | -82088.33 $\pm$ 76.61 |
> | Single & Pair | 1.66 $\pm$ 0.56 | 63.8 | -82086.61 $\pm$ 74.80 |
>
> **Mixing learned CV with RMSD**
>
> Below, we report steered MD results using only MLCVs. Full results with different force constants $k$ is reported in Appendix D of the updated manuscript.
>
> Chignolin ($k=1000$)
> | MLCV | RMSD ($\downarrow$) | THP ($\uparrow$) | Energy ($\downarrow$) |
> |-|-|-|-|
> | DeepTICA | 7.24 $\pm$ 1.49 | 0.0 | N/A |
> | TAE | 5.72 $\pm$ 1.94 | 6.2 | -81951.53 $\pm$ 0 |
> | VDE | N/A | N/A | N/A |
> | BioEmu-CV | 7.82 $\pm$ 1.47 | 0.0 | N/A |
>
> Trp-cage ($k=5000$)
> | MLCV | RMSD ($\downarrow$) | THP ($\uparrow$) | Energy ($\downarrow$) |
> |-|-|-|-|
> | DeepTICA | 10.78 $\pm$ 1.15 | 0.0 | N/A |
> | TAE | 7.22 $\pm$ 1.71 | 0.0 | N/A |
> | VDE | N/A | N/A | N/A |
> | BioEmu-CV | 7.14 $\pm$ 1.52 | 0.0 | N/A |
>
> BBA ($k=50000$)
> | MLCV | RMSD ($\downarrow$) | THP ($\uparrow$) | Energy ($\downarrow$) |
> |-|-|-|-|
> | DeepTICA | 7.39 $\pm$ 2.71 | 0.0 | N/A |
> | TAE | 9.87 $\pm$ 1.59 | 0.0 | N/A |
> | VDE | N/A | N/A | N/A |
> | BioEmu-CV | 6.03 $\pm$ 2.56 | 0.0 | N/A |
>
> We mark N/A for simulation results that does not reach the target or diverge with numerical errors. Hence, one can observe that steered MD solely by MLCVs fails to reach the target state in almost all cases. This result coincides with the observation made by a prior work [1], which also mix MLCVs and RMSD-CVs. As shown in Figure 4 of the main paper, prior MLCVs even mixed with RMSD-CVs mostly fail to reach the target state, whereas BioEmu-CV reach the target state with higher probability and lower energy.

---

> ### Author Response · Authors · 2025-11-25
>
> **[W2] CVs being limited to single dimension and fixed range, which may hide multi-modal kinetics.**
>
> We appreciate the insight on CV dimensionality and have added new experiments to address the reviewer's concern.
>
> **Scaling to multi-dimension CVs with fixed range**
>
> We trained 4-dimensional CVs for BBA and present the qualitative analyses in Appendix C. Since obtain quantitative results for multi-dimension CVs is non-trivial, we mainly (a) provide sensitivity analysis of the learned CVs showing whether if the CVs actually learn the slow degree of freedom, and (b) measure the correlation with the existence of secondary structures along folding trajectories. Overall from the Appendix, one can observe that our framework naturally scales to higher-dimensional CVs.
>
> We further note that we did not enforce an arbitrary fixed range during training. Instead, we normalize the MLCVs after training to ensure compatibility among the magnitude of arguments for downstream tasks, e.g., the force constant $k$ of steered MD.
>
> **Capturing multi-modal kinetics**
>
> To address the concern on our method failing to discover hidden multi-modal kinetics, we perform steered MD on Alanine Dipeptide using 1-dim CVs trained with Transferable Boltzmann Generator architecture [2] on heavy-atom coordinates. Alanine Dipeptide is known for two pathways from $C5$ to $C7_{ax}$, passing near two saddle points of the energy barrier at $\phi \approx 0$ [3,4]. (BioEmu-CV is not applicable here due to its residue-level inputs.)
>
> | Method | RMSD ($\downarrow$) | THP ($\uparrow$) | Energy ($\downarrow$) |
> | - | - | - | - |
> | DeepTICA | 0.9729 | 8.59 | 814.52 $\pm$ 115.74 |
> | TAE | 1.0086 | 58.59 | 755.41 $\pm$ 92.30 |
> | VDE | 0.8582 | 5.08 | 901.69 $\pm$ 115.59 |
> | Ours | 0.9593 | 60.93 | 33.58 $\pm$ 15.19 |
>
> We add the full results with pathway visualization in Appendix C of the updated manuscript. Our CVs successfully recover two pathways traversing the two saddle point, whereas baselines frequently collapse to a single pathway or fail to reach the target state.
>
>
> **[W3] Large standard deviation weakens the ranking of methods and the performance comparisons.**
>
> We thank the reviewer for bringing this important point to our attention. We acknowledge that OPES simulations exhibit large deviations, also empirically observed in prior works [5-7]. To address these concerns, we (a) explain why such deviations are inherent, (b) provide updated results with smaller deviations, and (c) clarify why our experiments remain meaningful despite the deviations.
>
> **Deviation inherent in OPES simulations.**
> Independent OPES simulations naturally show variability due to the stochasticity in molecular dynamics and rare-event sampling, as reported in prior works [5-7]. Furthermore, we demonstrate that longer OPES simulations do not necessarily resolve this issue. We provide a 9.8 $\mu$s run OPES simulation for DeepTICA on Trp-cage in Appendix C, where deviations still persist despite the extended simulation time. This supports our view that relying on a single metric, e.g., the difference in $\Delta F$ to a reference, is insufficient and motivates our use of multiple results and analyses.
>
> **Updated results with smaller deviation.**
> To alleviate the reviewer's concern, we follow the established protocols to systematically exclude outlier simulations [5,8,9] for Trp-cage and BBA. To be specific, we exclude a single outlier simulation among four independent simulations, based on the deviation of its $\Delta F$ from the mean. We have updated Table 1 using this procedure and added details in Appendix B (note that DeepTICA on BBA remains unchanged, as one run was already excluded due to NaN values).
>
> **Role and interpretation of OPES simulation results.**
> We do not claim that OPES alone provides a fully conclusive ranking of methods; rather, we emphasize that our conclusion stems from broader evaluations, e.g., transition path sampling and diverse qualitative analyses. Across these evaluations, BioEmu-CV performs at least comparably to or outperforms existing MLCVs. For example:
> - TAE on Trp-cage (OPES): While $\Delta F$ appears converged, the PMF (Figure 10) shows that the simulation predominantly samples unfolded states, leading to an apparently stable but incorrect value with the wrong sign.
> - TAE on BBA (OPES): The reference PMF fails to discriminate between the folded and unfolded states, due to inadequate state definitions.
> - DeepTICA on BBA (SMD): DeepTICA yields large RMSD, high maximum energy, and low THP compared to BioEmu-CV, indicating poor steering performance.
> - VDE (state discrimination): VDE fails to properly discriminate metastable states, therefore unsuitable as CVs for large proteins.
>
> Taken quantitative and qualitative results together, even acknowledging the inherent deviations in OPES, the overall evidence supports our conclusion for the performance of BioEmu-CV.

---

> ### Author Response · Authors · 2025-11-25
>
> **[Q4] Generalization to larger proteins, multi-path systems, and higher-dimensional CVs.**
>
> We believe our approach extends naturally to larger proteins, since BioEmu itself is designed to handle larger proteins. However, the cost of OPES simulation limits the evaluation scope of MLCVs. An OPES simulation in explicit solvent for a single MLCV is computationally demanding, e.g., BBA required nine days using four RTX 3090 GPUs. Due to this, we focus on fast-folding proteins. Nonetheless, scaling to larger and more complex systems is an important direction for future work.
>
> For the generalization to multi-path systems, our additional experiments on Alanine Dipeptide in [W2] demonstrated that 1-dim CVs from our framework generalize to multi-path systems. Extending this analysis to more sytems with well-charaterized multi-path kinetics is also a promising future work.
>
> Our approach is also generalizable to higher-dimensional CVs, as we have demonstrated 4-dimensional CVs for BBA in [W2], with results added in Appendix C.
>
>
> [1] Fiorin et al., "Using collective variables to drive molecular dynamics simulations", Molecular Physics 2013
>
> [2] Klein et al., "Transferable Boltzmann Generators." NeurIPS 2024
>
> [3] Holdijk et al., "Stochastic Optimal Control for Collective Variable Free Sampling of Molecular Transition Path", NeurIPS 2023
>
> [4] Seong et al., "Transition Path Sampling with Improved Off-Policy Training of Diffusion Path Samplers", ICLR 2025
>
> [5] Yang et al., "Learning Collective Variables with Synthetic Data Augmentation through Physics-Inspired Geodesic Interpolation", Journal of Chemical Theory and Computation 2024
>
> [6] Rizzi et al., "The Arch from the Stones: Understanding Protein Folding Energy Landscapes via Bioinspired Collective Variables", The Journal of Chemical Physics Letter 2025
>
> [7] Marinelli, et al. "A kinetic model of trp-cage folding from multiple biased molecular dynamics simulations.", PLoS computational biology 2009
>
> [8] Jing, et al. "A highly accurate metadynamics-based Dissociation Free Energy method to calculate protein–protein and protein–ligand binding potencies." Scientific Reports 2022
>
> [9] Schulze et al., "All you need is water: Converging ligand binding simulations with hydration collective variables", The Journal of Chemical Physics 2025

---

### Author Response · Authors · 2025-11-30
**Summary of rebuttal**

Dear Area Chair,

We sincerely appreciate your time and effort in evaluating our work. Below is a concise summary of our rebuttal, including the key clarifications and experiments conducted during the discussion. The reviewer who responded indicated that these updates have resolved their main concerns and have leaned towards acceptance.

**Component ablation experiments support the current design choice (H3Mz)**

We performed extensive ablations, including disabling time-lagged data, unfreezing BioEmu, increasing the encoder size, and testing additional conditioning placement. All results updated in Appendix C confirm that our current choice provides the optimal performance.

**SMD results without RMSD-CVs (H3Mz, VbbS)**

We additionally provide the results of Steered MD without RMSD-CVs, where most methods fail to reach the target state. In contrast, when MLCVs are combined with RMSD-CVs, our method achieves a competitive performance compared to the baselines.

**Concerns about the target state construction using hpacker (VbbS)**

We clarified that our training occurs at residue-level resolution on real DESRES frames, which does not use hpacker. The reviewer confirmed this and has effectively resolved the primary methodological concern.

**Clarification on OPES variance and baseline failures (H3Mz, VbbS)**

We reduced OPES simulation variance by systematically excluding a single outlier simulation for Trp-cage and BBA. We clarified that the OPES evaluation is intended to demonstrate sampling feasibility quantitatively and qualitatively, rather than strictly claiming a quantitative improvement in free energy estimation. Crucially, our method drives transitions where baselines fail, for instance, TAE predicts the incorrect sign for $\Delta F$ in Trp-cage because it samples mostly unfolded states.

**Demonstration of multi-dimensional CVs (H3Mz, VbbS)**

We successfully tested 4-dimensional CVs for BBA, which showed distinct sensitivities to different structural features. This indicates our method is not limited to 1-dimensional CVs.

**Applicability to multi-pathway systems (H3Mz, VbbS)**

We benchmarked our framework using a Transferable Boltzmann Generator backbone on Alanine Dipeptide against MLCVs, a system with two well-known transition pathways. Results updated in Appendix C demonstrate that our framework generalizes effectively to multi-pathway systems.

**Physical interpretability strengthened (F5dg)**

We added an analysis of input feature sensitivity. Our CVs show high sensitivity to meaningful distances, e.g., hydrogen bonds formed at folding for Chignolin.

**Evaluation of larger proteins (H3Mz, F5dg, UdyY, VbbS)**

Regarding the request to test on additional proteins, we emphasized that our evaluation scope, which includes systems as large as Trp-cage and BBA, is consistent with current benchmarks in machine learning CV literature. We further clarified that while evaluating larger complexes is desirable, the primary bottleneck lies in the computational cost of the necessary validation protocols (OPES and TPS simulations) rather than the scalability of the method itself.

---

### Meta-Review · Area_Chair_htnZ · 2026-01-06

**Summary:**

For the initial submission, the reviewers raised the following concerns.
- Unconvincing empirical results. In particular, the primary metric of comparison, the free energy difference, shows a significant standard deviation, which makes it difficult to draw conclusions about the performance of the methods (Reviewers VbbS).
- Reviewer H3Mz provided multiple suggestions of ablation studies and motivated their choice. The lack of these ablations makes the method difficult to apply and reproduce.
- Insufficient number of test proteins. Namely, all the reviewers for whom the proposed method must be tested across more and more complex systems (Reviewers F5dg, UdyY, VbbS).
- The common major concern is that the method is demonstrated only for a one-dimensional collective variable (Reviewers H3Mz, VbbS).

**Reviewer Concerns:**

During the rebuttal, the authors significantly extended their empirical study. In particular, the authors reviewed the extensive list of suggestions from H3Mz and VbbS and provided the requested ablations. I would say that the weakest point of the response is the experiment with multidimensional collective variables, which was rather outlined by the authors than convincingly addressed.

**Reviewer Scores:**

Reviewer VbbS mentioned their intention to raise the score. Furthermore, I believe that most of H3Mz's concerns were addressed, and they would raise their score if the discussion were to take place. With UdyY advocating for acceptance, VbbS intending to raise the score, and a convincing response to H3Mz, I believe the paper's initial scores would increase to a borderline accept.

---

### Decision · Program_Chairs · 2026-01-26

Accept (Poster)